# Chlorantraniliprole Enhances Cellular Immunity in Larvae of *Spodoptera frugiperda* (Smith) (Lepidoptera: Noctuidae)

**DOI:** 10.3390/insects15080586

**Published:** 2024-08-01

**Authors:** Qingyan Liu, Xiaoyue Deng, Liuhong Wang, Wenqi Xie, Huilai Zhang, Qing Li, Qunfang Yang, Chunxian Jiang

**Affiliations:** 1College of Agronomy, Sichuan Agricultural University, Chengdu 611130, China; qingyanliu1@163.com (Q.L.); 15928708807@163.com (X.D.); wlhangiuong@163.com (L.W.); wenqixie339@163.com (W.X.); huilai_zhang@126.com (H.Z.); liqing8633@sicau.edu.cn (Q.L.); lmk94811@163.com (Q.Y.); 2Emeishan Agricultural and Rural Bureau, Emeishan 614200, China

**Keywords:** cellular immunity, chlorantraniliprole, *Spodoptera frugiperda*, hemocytes

## Abstract

**Simple Summary:**

Insect innate immunity comprises cellular and humoral reactions and is crucial for combating pathogenic infections. Cellular immunity, which is executed rapidly by hemocytes, encompasses phagocytosis, nodulation, and encapsulation. Some studies have indicated that pesticides may affect the immune function of insects, but different insects respond differently to pesticides. This study investigated the impact of chlorantraniliprole on hemocytes and cellular immunity in larvae of the fall armyworm (FAW), *Spodoptera frugiperda*. Our results show that the LD_10_, LD_20_, and LD_30_ of chlorantraniliprole could increase the spreading ratio of plasmatocytes and the areas of the cytoskeletons in hemocytes, resulting in enhanced cellular immune function in FAW larvae. This provides a foundation for further exploration of the cellular immunity mechanisms of FAW.

**Abstract:**

The innate immunity of insects encompasses cellular and humoral defense mechanisms and constitutes the primary defense against invading microbial pathogens. Cellular immunity (phagocytosis, nodulation, and encapsulation) is primarily mediated by hemocytes. Plasmatocytes and granulocytes play an important role and require changes in the cytoskeletons of hemocytes. However, research investigating the immunological impacts of insecticides on the fall armyworm (FAW), *Spodoptera frugiperda*, remains scarce. Therefore, we conducted a study to investigate the effects of chlorantraniliprole exposure on cellular immunity in FAW larvae. Our findings revealed the presence of five types of hemocytes in the larvae: prohemocytes, plasmatocytes, granulocytes, oenocytoids, and spherulocytes. The LD_10_, LD_20_, and LD_30_ of chlorantraniliprole affected both the morphology and total count of some hemocytes in the larvae. Moreover, larvae exposed to chlorantraniliprole showed increased phagocytosis, nodulation, and encapsulation. To determine the mechanism of the enhanced cellular immunity, we studied plasmatocytes in the spread state and the cytoskeleton in hemocytes. It was found that the spreading ratio of plasmatocytes and the areas of the cytoskeletons in hemocytes were increased after chlorantraniliprole treatment. These results suggest that exposure to chlorantraniliprole results in an enhanced immune response function in FAW larvae, which may be mediated by cytoskeletal changes and plasmatocyte spreading. Consequently, this study provides valuable insights into the cellular immune response of FAW larvae to insecticide exposure.

## 1. Introduction

Insects, similar to mammals, inhabit environments teeming with a diverse array of pathogens [1]. To resist these pathogens, insects have developed an array of defense mechanisms. Primarily, the integument serves as the initial physical barrier, directly preventing pathogen invasion [2]. Subsequently, the immune system plays a crucial role when the integument is compromised [3,4]. Moreover, innate immunity encompasses both humoral and cellular defense responses [5], and cellular immunity is mediated by hemocytes. The hemocytes found in Lepidopterans include prohemocytes, plasmatocytes, granulocytes, oenocytoids, and spherulocytes [6]. Among these, plasmatocytes and granulocytes are the principal hemocyte types capable of adhering to foreign surfaces, typically constituting over 50% of hemocytes [7,8]. However, hemocyte counts in insects vary depending on the developmental stage, physiological conditions, and infection status [4]. For instance, following injection of *Beauveria bassiana* (Balsamo) conidia into *Conogethes punctiferalis* (Guenée), significant decreases in the total and differential hemocyte counts were observed over time [9]. Similarly, the total hemocyte count in *Galleria mellonella* L. infected with *Actinobacillus pleuropneumoniae* Pohl et al. fluctuated during the infection [10].

Hemocytes exhibit various immune functions and orchestrate immune responses upon recognizing foreign substances as non-self in insects. The most prominent immune responses mediated by hemocytes are phagocytosis, nodulation, and encapsulation [8,11]. Phagocytosis is a widely conserved defense response to a diversity of targets, including bacteria, fungal spores, and small abiotic particles [12,13]. Nodulation refers to multicellular aggregates that entrap large numbers of bacteria [14]. Additionally, encapsulation refers to the binding of hemocytes to larger targets, such as parasites, protozoa, nematodes, or abiotic particles [15]. Specifically, in Lepidopteran insects, the nodulation and encapsulation primarily involve the participation of plasmatocytes and granulocytes [16]. Consequently, plasmatocytes and granulocytes play pivotal roles in cellular immunity. For instance, infection of *Spodoptera frugiperda* (Smith) with *Metarhizium rileyi* (Farlow) led to significant reductions in the total counts of hemocytes, plasmatocytes, and granulocytes, which consequently affected the cellular immune response [17]. Furthermore, plasmatocytes require filopodia formation for functionality, which entails morphological alterations [18]. Cell morphology and motility are controlled by the cytoskeleton of the cell, which is mainly composed of actin filaments. These filaments reshape cell morphology by polymerizing and depolymerizing [19,20].

Insect immune responses can be influenced by various external factors, such as temperature, population density, pathogens, and pesticide exposure [21,22,23]. Previous research has identified two main aspects regarding the impacts of pesticides on insect immunity. Certain pesticides can diminish insect immune responses, thereby increasing susceptibility to pathogen infections [24]. For instance, Delpuech et al. [25] reported that two insecticides, dieldrin and endosulfan, decreased the capacity of *Drosophila melanogaster* (Meigen) to encapsulate parasitoid eggs and impaired the fitness of larvae. Neonicotinoids, such as thiacloprid, imidacloprid, and clothianidin, have been shown to negatively affect hemocyte density, the encapsulation response, and antimicrobial activity in honeybees, thus diminishing their immunocompetence and potentially making them more prone to parasites and pathogens [26]. Similarly, the neonicotinoid clothianidin had a negative impact on immune responses, which can boost the proliferation of honeybee parasites and pathogens [27,28]. Conversely, pesticides can stimulate and enhance insect immune responses, thereby bolstering resistance to pathogens [29]. Dubovskiy et al. [30] demonstrated that sublethal doses of organophosphorus insecticides significantly increased the numbers of hemocytes and the encapsulation activity in *G. mellonella* and *Leptinotarsa decemlineata* (Say). Additionally, when *G*. *mellonella* larvae were infected with *Bacillus thuringiensis* Berliner, their cellular and humoral immune responses were enhanced, enabling them to resist its effects [31].

Insects exhibit complex and multifaceted immune systems, which provide protection against a wide range of pathogens, contributing to their extensive distribution across diverse ecosystems [32]. The fall armyworm (FAW), *Spodoptera frugiperda* (Smith) (Lepidoptera: Noctuidae), has emerged as a notable pest that significantly impacts various crops, notably maize and cotton, across the Americas [33]. Its presence was first documented in China in January 2019 [34], and its substantial impact on crop production was attributed to its varied feeding habits, strong migratory tendencies, severe infestations, and high fecundity [35,36,37]. Currently, chemical pesticides are one of the most effective ways to control the outbreak of FAW in the short term due to their high economic efficiency [38].

Chlorantraniliprole, a systemic insecticide with a broad spectrum of activity against Lepidopteran pests [39], is classified as an essential emergency measure for the control of FAW in the context of integrated pest management (IPM) [40]. This chemical functions by activating ryanodine receptors (RyRs), which leads to uncontrolled calcium release in muscle cells, resulting in continuous muscle contraction and, ultimately, the death of the insect [41]. In addition to their short-term lethal effects, insecticides may also exhibit long-term sublethal effects on exposed pests due to their volatilization and degradation, as well as improper application under field conditions [42,43,44]. For example, exposing FAW to LC_10_ or LC_30_ chlorantraniliprole prolonged the larval duration, decreased the mean weights of the larvae and pupae, and reduced both the pupation rate and the adult emergence rate [45]. Zhang et al. [46] indicated that both emamectin benzoate and chlorantraniliprole prolonged the development of the F_0_ generation of FAW, and fecundity was reduced with increasing insecticide concentrations. Specifically, chlorantraniliprole prolonged the pre-adult and adult stages of FAW at LC_10_ and LC_25_, respectively, which significantly improved fecundity. The effects of different pesticides on insect immunity exhibit substantial variation. Although chlorantraniliprole is frequently employed for pest control, its impact on FAW immunity remains unclear. Therefore, this study aims to investigate the effects of chlorantraniliprole exposure on the cellular immunity of FAW larvae.

In this study, we characterized the hemocyte types present in 4th instar larvae of FAW. We examined the impacts of chlorantraniliprole exposure on morphology and abundance of hemocytes, as well as phagocytosis, encapsulation, and nodulation in FAW larvae. Additionally, we investigated its influence on the spreading ratio of plasmatocytes and the area of the cytoskeleton in hemocytes. These findings contribute to our comprehension of the cellular immune response in FAW larvae and provide a basis for understanding the effects of insecticides on insect immune responses.

## 2. Materials and Methods

### 2.1. Insects and Insecticides

The FAW was obtained from Luzhou City, Sichuan Province, China, in July 2020. The larvae were fed an artificial diet (Appendix A) under laboratory conditions for several generations for this study. Adults were fed a 10% honey water solution, 1st and 2nd instar larvae were fed in a square box, and 3rd instar larvae were individually placed in finger tubes. The rearing conditions were 27 ± 1 °C, 70% ± 5% relative humidity (RH), and a 10:14 h photoperiod. Additionally, all treated larvae were fed under these same conditions. On the first day of the 4th instar, the larvae were weighed individually, and those weighing approximately 0.025 ± 0.005 g were selected for all experiments.

Chlorantraniliprole (95% TC, DuPont, Beijing, China) was dissolved in acetone (KESHI, Chengdu, China) to make a 1 mg/L stock solution.

### 2.2. Larval Toxicity Bioassay

The drip method was used as described by Lu et al. [47] with slight modifications. Briefly, the stock solution of chlorantraniliprole was diluted to serial concentration gradients (31.25, 62.5, 125, 250, and 500 mg/L) with a 0.1% Triton X-100 (Beyotime, Chengdu, China) aqueous solution [45]. Then, 0.2 µL of each diluted chlorantraniliprole concentration was individually dropped onto the pronotum of each 4th instar larva using a micro-syringe (Shanghai Anting Microsampler Factory, Shanghai, China). The weights of the tested larvae were controlled. Thus, each group of larvae was exposed to a certain dose of chlorantraniliprole, with the doses being 0.25, 0.5, 1, 2, and 4 μg/g, respectively. Then, we used the mortality of the larvae treated with the chlorantraniliprole doses for analysis. The control groups (CK) were treated with 0.2 µL of a 0.1% Triton X-100 aqueous solution. Each treatment was performed 3 times, with 15 larvae per replicate. The treated larvae were fed an artificial diet during the trial. After 24 h of rearing in a single tube, mortality data were recorded. The larvae were considered dead when they remained immobile after being touched with a soft paintbrush.

### 2.3. Hemolymph Collection

Larvae were treated with chlorantraniliprole at doses of LD_10_, LD_20_, and LD_30_. After treatment for 6, 12, 18, 24, and 48 h, the larvae were placed on ice for immobilization for 15 min. The abdominal prolegs were then sterilized by swabbing three times with 75% ethanol and subsequently punctured using an insect needle. Hemolymph was collected into a pre-cooled PCR tube using an Eppendorf pipette (Research plus, Eppendorf, Hamburg, Germany). Hemolymph was only collected once from each larva. The hemolymph was transferred to an Eppendorf tube for subsequent experiments.

### 2.4. Hemocyte Assays

Next, 5 μL of hemolymph was collected and spotted on a glass slide. Once dried, the hemolymph was fixed with 100 µL of methanol (Xilong Science, Chengdu, China) for 4 min. The slide was then stained with 3–4 drops of Giemsa–Wright staining solution (BBI, Shanghai, China) for 8 min at 25 °C. After staining, the slides were rinsed with distilled water and covered with a coverslip. The hemocyte types and morphology were observed under a positive fluorescence microscope (Axio ImagerZ2, Zeiss, Oberkochen, Germany). The hemocyte types were identified according to the description of Li et al. [9] under optical and differential interference contrast (DIC) microscope, which are two modes of positive fluorescence microscopy. Moreover, morphologic changes in the hemocytes were observed and documented according to the description by Mao et al. [48].

### 2.5. Total Hemocyte Counts

Next, 20 μL of hemolymph was diluted 1:2 in a saline solution (137.78 mM NaCl, 8.51 mM KCl, 3.87 mM CaCl_2_, 1 mM MgCl_2_, and 10 mM HEPES (pH 7.0)). The total hemocyte counts were quantified in an improved Neubauer hemocytometer chamber (Shanghai Qijing Biochemical Instrument Co., Ltd., Shanghai, China) using a biological microscope (CX22, Olympus, Tokyo, Japan). Each treatment was performed 10 times, with 10 larvae per replicate. The total number of hemocytes was calculated according to the following formula: total number of hemocytes (cells/mL) = (5N × 10^4^)/dilution ratio (N represents the number of cells in the four corners and five middle compartments of the hemocytometer) [49].

### 2.6. Phagocytosis Assay

Hemolymph was extracted from the treated larvae and diluted (1:5) with a saline solution, and 20 µL of the diluted hemolymph was dropped onto a glass slide. The slide was incubated in a humidor for 30 min at 28 °C. Then, FITC-conjugated *Escherichia coli* (Escherich; 4 × 10^5^ cells/mL) was added to the slide. The slide was then placed in the humidor and incubated at 28 °C for 1 h. Subsequently, 100 µL of a 0.4% Trypan Blue Stain solution (Solarbio, Beijing, China) was added to the slides. Each slide was stained for 5 min and washed three times with 1 × PBS buffer (pH 7.2–7.4; Solarbio, Beijing, China). Finally, the slides were randomly observed under the positive-fluorescence microscope (Axio ImagerZ2, Zeiss, Oberkochen, Germany). Each treatment was performed 3 times, with 5 larvae per replicate. A minimum of 150 hemocytes were analyzed for each treatment. After the Trypan Blue Stain solution was added, green fluorescence was still observed in the hemocytes, but the extracellular fluorescence was quenched. Intact hemocytes with green fluorescence were determined to be phagocytic hemocytes. The phagocytosis ratio was calculated according to the following formula: phagocytosis ratio = (number of hemocytes in which phagocytosis occurred/total number of hemocytes) × 100% [50].

### 2.7. Nodulation Assay

Nodules are black sand or black spots that flow with the hemolymph or adhere to the midgut and fat body in insects [51]. To examine nodulation, the treated larvae were placed on ice for immobilization for 15 min. Then, 3 µL of *E. coli* (1 × 10^5^ cells/mL) was injected into the tested larvae, which were then incubated for 3 h at 28 °C. The abdomen of larvae was opened by cutting the epidermis from head to tail. Based on the method of Sadekuzzaman et al. [52], with slight modifications, black nodules were counted on the fat body and midgut under a stereomicroscope (SZX16, Olympus, Tokyo, Japan). The diameter of each nodule was measured using Image J 1.54f software. Each 30 μm was counted as one nodule (for example, 1–30 μm was one nodule, 31–60 μm was two nodules, 61–90 μm was three nodules, and so on) [53]. Each treatment was performed 3 times, with 10 larvae per replicate.

### 2.8. Encapsulation Assay

Encapsulation was performed based on the method of Ni et al. [54], with some modifications. Briefly, Sephadex A-25 chromatography beads were added to 2 mL of a 0.1% Congo red dye solution to stain for 8 h. After staining, the excess dye solution was washed with distilled water. The stained beads were added to 1 mL of a saline solution to make a saline suspension of Congo red-stained DEAE-Sephadex A-25 chromatography beads, which was stored at 4 °C. The treated larvae were placed on ice for immobilization for 15 min. Then, each larva was injected with 10–20 Congo red-stained DEAE-Sephadex A-25 chromatography beads. After injection, the larvae were incubated for 24 h at 28 °C. The tested larvae in each group were dissected under a stereomicroscope (SZX16, Olympus, Tokyo, Japan). All beads inside the tested larvae were removed and observed under the positive-fluorescence microscope (Axio ImagerZ2, Zeiss, Oberkochen, Germany). The thickness of the encapsulation (L) and the bead diameter (D) were measured using Image J 1.54f software. According to the method of Hu et al. [55], with some modifications, the encapsulation of the beads was classified into 13 grades, as follows: grade 1: the beads were not encapsulated or only had small numbers of cells adhering to them, grade 2: the beads were not completely encapsulated, grade 3: the beads were completely encapsulated, but 0 < L/D ≤ 0.1, grade 4: 0.1 < L/D ≤ 0.2, and so on, with L/D > 1 for grade 13. The encapsulation index was = Σ(i × Pi), where i represents the encapsulation level of the beads (1, 2, …, 13) and Pi represents the proportion of i beads compared to the total number of beads. Each treatment was repeated 5 times, with 5 larvae per replicate.

### 2.9. Plasmatocyte-Spreading Assay

Here, 20 μL of hemolymph was collected from treated larvae and diluted (1:5) with a saline solution. Subsequently, the diluted hemolymph was applied to a slide, resulting in three slices for each treatment. The slices were then placed in a humidor, incubated at 28 °C for 30 min, and observed under the positive-fluorescence microscope (Axio ImagerZ2, Zeiss, Oberkochen, Germany). According to the method of Huang et al. [56], plasmatocytes that were spindle-shaped or irregularly shaped were counted as spread. Each treatment was repeated 3 times, with 5 larvae per replicate. For each treatment group, a minimum of 180 hemocytes was analyzed. The spreading ratio of plasmatocytes was calculated as: (number of plasmatocytes that have been spread/total number of plasmatocytes) × 100%.

### 2.10. Hemocyte Cytoskeleton Assay

To evaluate the cytoskeletal areas in hemocytes of treated larvae, diluted hemolymph (1:5) was applied to a slide. The slices were placed in the humidor and incubated at 28 °C for 30 min. According to the method of Liu et al. [57], with some modifications, hemocytes were slowly washed with PBS, fixed in 4% formaldehyde for 10 min, and slowly washed with PBS. Then, each slide was permeabilized with 0.2% Triton X-100 (200 μL) for 10 min. After slowly washing with PBS, the hemocytes were stained with 100 μL of phalloidin-FITC (Solarbio, Beijing, China) for 2~3 h. Finally, the slides were gently washed with PBS and imaged using the positive-fluorescence microscope (Axio ImagerZ2, Zeiss, Oberkochen, Germany). Each treatment was repeated 10 times, with 5 larvae per replicate. For each treatment group, a minimum of 100 hemocytes were analyzed. To exclude interference from broken cells and other impurities, the cytoskeletal areas of hemocytes larger than 100 μm^2^ were measured using Image J 1.54f software.

### 2.11. Statistical Analyses

All data analysis was performed using SPSS 27.0.1 software (SPSS Inc., Chicago, IL, USA). In the larval toxicity bioassay, the LD_10_, LD_20_, LD_30_, and LD_50_ values with 95% confidence limits, chi-square value, and slopes of the log-dose probit mortality lines were calculated via probit regression analysis. The Shapiro–Wilk and Levene’s tests were used to verify the assumptions of normality and homogeneity of variance before analysis. If the data met the assumptions of normality and homogeneity of variance, we conducted a one-way analysis of variance followed by an LSD multiple comparison analysis of significant differences. If the data did not meet the assumptions of normality and homogeneity of variance, the nonparametric Kruskal–Wallis test of variance was used, followed by multiple-comparison tests for the analysis of significant differences. *p*-values less than 0.05 were considered statistically significant. The fluorescence images were analyzed using Image J 1.54f software, and the data were plotted using GraphPad Prism 8.0.2 software (GraphPad Inc., San Diego, CA, USA).

## 3. Results

### 3.1. Larval Toxicity Bioassay

The chlorantraniliprole toxicity results for the 4th instar larvae of FAW are shown in Table 1. The results of the bioassay demonstrated the high insecticidal efficacy of chlorantraniliprole against 4th instar larvae of the FAW.

### 3.2. Hemocyte Types

As shown in Figure 1, five types of hemocytes were identified in FAW larvae: prohemocytes, plasmatocytes, granulocytes, oenocytoids, and spherulocytes. Of all the cells observed, prohemocytes were small and round, with smooth surfaces. The nucleus-to-cytoplasm ratio was high, which helped to differentiate these cells from other types of hemocytes (Figure 1a,b). Plasmatocytes with variable sizes were highly polymorphic. Most presented a spindle shape, while a few showed oval or spherical shapes under the light microscope. These cells were characterized by numerous irregular processes, such as lamellipodia and filopodia, and they showed a large, centrally localized polymorphic or rounded nucleus (Figure 1c,d). Granulocytes showed circular or oval profiles with highly variable sizes and contained several refractive inclusions. These cells were characterized by the inclusion of several dense and structured granules. The plasma membranes of these cells were irregular, presenting projections as filopodia surrounding the cells (Figure 1e,f). Oenocytoids appeared as large cells that were rather regular in shape, and had low nucleus-to-cytoplasm ratios. The nuclei of oenocytoids often had eccentric locations. The cytoplasm was homogenous, and a few organelles, such as vesicles and small electron-dense granules, could be observed (Figure 1g,h). Spherulocytes were rounded cells that contained a small number of large inclusions (spherules), which caused the cells to adopt irregular shapes (Figure 1i,j).

### 3.3. Hemocyte Morphology

After the larvae were treated with chlorantraniliprole, some hemocytes exhibited a range of damage, including vacuolation, pyknosis, degranulation, nucleus deformation, nucleus apoptosis, and cell membrane deformation. Upon exposure to varying doses, cytoplasmic vacuolization was observed in some hemocytes (Figure 2a–c). Hemocyte pyknosis, characterized by overall cytoplasm and nucleus reductions along with darkening, occurred after 6 h of exposure (Figure 2d–f). Some granulocytes were degranulated, and hemolysis in the nucleus was observed (Figure 2g–i). After 48 h of treatment at each dose, hemocytes displayed nucleus deformation with irregular edges (Figure 2j–l). Some hemocytes underwent nucleus apoptosis, leading to the formation of multiple spherical apoptotic vesicles (Figure 2m–o). The hemocyte membrane displayed protrusions, and fully folded circular ruffles appeared after the larvae were treated with different doses (Figure 2p–r).

### 3.4. Total Hemocyte Counts

After treatment with chlorantraniliprole for 48 h, the total numbers of hemocytes in the FAW larvae in both the control and treatment groups initially increased and then decreased over time. At 12 h, the total numbers of hemocytes in the LD_20_ and LD_30_ groups increased significantly compared to the control group (*F* = 25.402, *p* < 0.0001). At 24 h, the numbers of hemocytes in the LD_10_ and LD_20_ groups were significantly lower than in the control group (*F* = 23.701, *p* < 0.0001). At 48 h, the numbers of hemocytes in the LD_10_, LD_20_, and LD_30_ groups were significantly lower than that in the control group (*F* = 15.490, *p* = 0.001). However, at 6 and 18 h, there were no significant differences in the hemocyte counts among the LD_10_, LD_20_, and LD_30_ groups and the control group (Figure 3).

### 3.5. Phagocytosis Assay

As shown in Figure 4, at 24 h, the LD_10_, LD_20_, and LD_30_ of chlorantraniliprole caused the greatest increases in the phagocytosis ratio during the treatment (*F* = 12.434, *p* < 0.01). At 6, 12, 18, and 48 h, there were no significant differences observed in the phagocytosis ratio among the LD_10_, LD_20_, and LD_30_ groups and the control group. However, at 18 h, the phagocytosis ratio was higher in the LD_10_, LD_20_, and LD_30_ groups than in the control group and increased with the dose, but this was not significant (Figure 4).

### 3.6. Nodulation Assay

The results indicated that varying doses of chlorantraniliprole promoted nodule formation in the hemocytes of FAW larvae to different extents. Notably, the LD_10_ and LD_20_ groups exhibited the most significant effects at 48 h (Figure 5). At 12 and 18 h, the numbers of nodules in the LD_20_ and LD_30_ groups were significantly higher than that in the control group (*F* = 6.360, *p* = 0.005; *F* = 28.447, *p* < 0.0001). Furthermore, the numbers of nodules at 18 h in the LD_10_, LD_20_, and LD_30_ groups increased proportionally with the dose. At 24 h, both the LD_10_ and LD_30_ groups had significantly more nodules compared to the control group (*F* = 6.397, *p* = 0.005). Finally, at 48 h, the numbers in the LD_10_ and LD_20_ groups were significantly higher than that in the control group (*F* = 80.176, *p* < 0.0001), with the LD_10_ group exhibiting the most significant increase in nodule formation. However, at 6 h, no significant differences were observed in the numbers of nodules in the LD_10_, LD_20_, and LD_30_ groups compared to the control group.

### 3.7. Encapsulation Assay

Within 48 h, different doses of chlorantraniliprole promoted the encapsulation of hemocytes in FAW larvae to varying degrees (Figure 6). At 18, 24, and 48 h, the encapsulation index increased significantly in the LD_10_, LD_20_, and LD_30_ groups compared to the control group (*F* = 15.752, *p* = 0.001; *F* = 150.147, *p* < 0.0001; *F* = 136.270, *p* < 0.0001). Notably, the encapsulation index increased with the dose at 18 h. At 24 and 48 h, the LD_30_ group had the greatest increases in the encapsulation index. At 6 h, the encapsulation index significantly increased in the LD_20_ group compared to the control group (*F* = 8.831, *p* = 0.039). At 12 h, the encapsulation index in the LD_30_ group was significantly higher than that in the control group (*F* = 4.259, *p* = 0.022).

### 3.8. Plasmatocyte-Spreading Assay

The obtained results indicated that exposure to chlorantraniliprole for 48 h could enhance the spreading ratio of plasmatocytes in FAW larvae. As shown in Figure 7, the spreading ratios of the plasmatocytes were highest at 18 h. Except for the LD_30_ group at 12 h, the spreading ratios of the plasmatocytes in the LD_10_, LD_20_, and LD_30_ groups were significantly different compared with the control group at 6 h (*F* = 77.174, *p* < 0.0001), 12 h (*F* = 29.014, *p* < 0.0001), 18 h (*F* = 146.514, *p* < 0.0001), 24 h (*F* = 13.966, *p* = 0.002), and 48 h (*F* = 16.736, *p* = 0.001). Furthermore, the spreading ratios of the plasmatocytes decreased progressively with increasing doses at each tested time. Among all tested times, the LD_10_ group had the greatest increase in the spreading ratio of the plasmatocytes.

### 3.9. Hemocyte Cytoskeleton Assay

As shown in Figure 8, treatment with various doses of chlorantraniliprole could promote the expansion of the areas of the cytoskeletons in hemocytes of FAW larvae. At 6 h, the areas of the cytoskeletons in the hemocytes in the LD_10_*,* LD_20_*,* and LD*_3_*_0_ group*s* were significantly greater than that in the control group (*F* = 6.884, *p* = 0.001) and increased with the dose. At 12 h, the areas of the cytoskeletons in the hemocytes in the LD_20_ and LD_30_ groups significantly increased compared to the control group (*F* = 5.653, *p* = 0.003). At 18 h, the LD_20_ group had a significantly greater cytoskeleton area than that the control group (*F* = 7.995, *p* = 0.046). At 24 h, the areas of the cytoskeletons in the hemocytes in the LD_10_ and LD_30_ groups were significantly greater than that in the control group (*F* = 19.534, *p* < 0.0001). At 48 h, the area of the cytoskeleton in the LD_10_ group was significantly greater than that in the control group (*F* = 8.715, *p* = 0.033). Furthermore, the cytoskeleton in the control group was spindle-shaped, and its area was significantly smaller than those in the treatment groups (Figure 9A). With LD_30_ chlorantraniliprole exposure, the cytoskeleton extended in all directions and became a rectangle (Figure 9B).

## 4. Discussion

Innate immunity serves as the primary initiator of the immune response against pathogenic invaders, forming the initial line of defense in the host [58]. Pesticides can influence the immune response of insects, with various types of pesticides having different mechanisms to affect the immune response [24]. This study investigated the impact of chlorantraniliprole exposure on cellular immunity in FAW larvae.

Insect cellular immunity is primarily mediated by hemocytes [16]. Due to the extensive diversity among insect species, hemocyte types exhibit considerable variation, even within the same species, depending on environmental conditions and developmental stages [59]. Among Lepidopterans, five primary hemocyte types have been commonly identified: prohemocytes, plasmatocytes, granulocytes, oenocytoids, and spherulocytes [2,7,50]. Li et al. [60] also observed these five types of hemocytes in FAW larvae, and this study corroborated their findings. Furthermore, our experiment showed the presence of certain cells that resembled the vermiform cells found in *Mythimna unipuncta* (Haworth). These cells contained small, dense granules but did not seem to be involved in capsule formation [61]. Their morphology was similar to the spread state of the plasmatocytes in our experiment, whose lengths reached approximately 80 μm. However, the diameters of the plasmatocytes in this experiment did not exceed 40 μm. Due to their extremely limited numbers, further research is needed to identify these cells.

Pesticide exposure can adversely affect the morphology of hemocytes in insects [62,63]. This study found vacuolization, cell membrane irregularities, nucleus deformation, and nucleus apoptosis in some hemocytes from each treatment group. Similar anomalies were observed in *Bombus terrestris* L., where vacuolization, cell membrane irregularities, and alterations in cell shape were detected in prohemocytes, plasmatocytes, granulocytes, spherulocytes, and oenocytoids after 48 h of thiamethoxam exposure [64]. These findings were consistent with those of Wang et al. [65], who reported pyknosis, vacuolization, and deformation of hemocytes following injection of azadirachtin into *Oxya chinensis* (Thunberg). Sublethal hexaflumuron exposure caused plasmatocyte filopodia to contract and shorten, caused granulocytes to compact with a loss of cytoplasmic projections, and changed the morphology and structure of granulocytes in larvae of the armyworm, *Mythimna separata* (Walker) [63]. These studies have demonstrated that chemical exposure can be toxic to hemocyte morphology, which is essential for their function. Morphological changes can impair the ability of hemocytes to combat illnesses. For example, granulocytes have been shown to actively produce various sticky nets from their plasma membranes, which they use to gather other hemocytes and to implement cellular immune responses [66]. Additionally, plasmatocytes can rearrange their cytoskeleton to surround invading pathogens with pseudopodia or filopodia [12]. Therefore, alterations in hemocyte morphology may adversely affect their functionality within the cellular immune response of FAW larvae.

The hemocyte count in insects is influenced by a variety of factors, including the specific molecules involved, the doses administered, and the particular species considered [16]. A study examining the effects of different sublethal doses of imidacloprid on the total hemocyte counts of *G. mellonella* revealed a decrease in hemocytes in the experimental group compared to the control group [67]. In this study, the total hemocyte counts fluctuated with the treatment time after the FAW larvae were treated with chlorantraniliprole. This phenomenon may be attributed to the initial resistance of the larvae to organismal damage shortly after treatment, which boosted their immunity by increasing the number of hemocytes. However, over a longer duration of treatment, some hemocytes participated in cellular immunity, resulting in a lower hemocyte count in the treatment group compared to the control group. Moreover, different types of hemocytes in insects exhibit varying responses to stress conditions caused by different insecticides [62]. In larvae of the armyworm *M. separata*, sublethal hexaflumuron exposure decreased the number of granulocytes and increased the number of plasmatocytes but had few effects on the counts of spherulocytes, oenocytoids, and prohemocytes [63]. The numbers of prohemocytes and plasmatocytes increased but granulocytes declined when the stingless bee, *Melipona quadrifasciata* Lepeletier, was exposed to imidacloprid [68]. Among the hemocyte types, plasmatocytes and granulocytes are considered major contributors to cellular immunity [7]. Fluctuation in the numbers of these two cell types under insecticide stress may impact cellular immunity. For example, the reduced numbers of granulocytes and plasmatocytes in FAW injected with *M. rileyi* blastospores may have contributed to their compromised capacity for encapsulation and nodulation [17]. However, other hemocyte types likely interact with plasmatocytes and granulocytes, contributing to the overall immune response [69]. The specific dynamics of each type of hemocyte in this study remain unclear and require further investigation.

Cellular defense functions, which are crucial for survival of insects, include a range of immune responses, such as phagocytosis, encapsulation, and the formation of hemocyte nodules [54]. Several studies have reported the ability of insecticides to affect cellular immunity in insects. For instance, Brandt et al. [26] indicated that thiacloprid and imidacloprid reduced the encapsulation reactions of honeybees at both laboratory and field concentrations. However, Dubovskiy et al. [30] demonstrated that the LC_10_ and LC_50_ of an organophosphorus insecticide treatment significantly enhanced the encapsulation of both the wax moth, *G. mellonella*, and the Colorado potato beetle, *L. decemlineata*. In this experiment, different doses of chlorantraniliprole promoted effects on the phagocytosis, nodulation, and encapsulation in FAW larvae. These results underscore the complex effects that different insecticides can have on the immune systems of various insects.

The morphological conversion of plasmatocytes from a non-spread state to a spread state is essential for capsule formation [15]. Hu et al. [55] reduced the spreading ratio of plasmatocytes using RNAi, resulting in a decrease in encapsulation of *Helicoverpa armigera* (Hübner). In this study, all treatment groups exhibited significant increases in the spreading ratios of plasmatocytes compared to the control group, with the LD_10_ treatment group showing the most pronounced effect. It was challenging to precisely differentiate the adhesive states of granulocytes. While we did not calculate the spreading ratios of granulocytes in this study, we cannot exclude the possibility that chlorantraniliprole may also influence granulocyte spreading behavior.

Plasmatocytes’ spread involves cytoplasmic morphological changes that allow for cell spreading through filopodia extension, a physiological process dependent on cytoskeletal rearrangement [70]. The cytoskeleton plays important roles in various essential cellular processes, including cell migration and motility, cell division, and the establishment and maintenance of cell and tissue architecture [20]. Previous research on *Lacanobia oleracea* L. parasitism by the ectoparasitic wasp, *Eulophus pennicornis* (Nees), showed that the cytoskeletons of host hemocytes were disrupted, leading to inhibition of encapsulation in vivo [71]. In addition, F-actin polymerization in hemocytes, especially in plasmatocytes, was significantly reduced, resulting in the hindrance of plasmatocyte spreading and a significant decrease in the chromatography bead encapsulation ratio, when integrin β1 expression in *Ostrinia furnacalis* (Guenée) hemocytes was inhibited by RNAi [72]. Thus, the cytoskeletons in hemocytes can influence cellular immunity. Regarding the insecticide, chlorantraniliprole can induce the release of Ca^2+^ from the intracellular calcium pool by activating the insect ryanodine receptor (RyR), leading to muscle paralysis and eventual death [41]. According to reports by Ahmed and Kim [70], inhibition of Ca^2+^ flux significantly impaired the hemocyte spreading and the subsequent cellular immune response (phagocytosis). They indicated that PGE_2_ mediates hemocyte spreading via the cAMP signal to activate aquaporin and via the Ca^2+^ signal to activate actin cytoskeletal rearrangement. Therefore, we speculate that chlorantraniliprole may mediate Ca^2+^ flux in hemocytes and may ultimately affect cytoskeletal rearrangements. However, further studies are needed to investigate the changes in Ca^2+^ flux in hemocytes after exposure to chlorantraniliprole and the effects of Ca^2+^ flux on the immunity of FAW larvae. 

Our findings indicate that exposure to chlorantraniliprole can enhance the cellular immune response in FAW larvae. This heightened cellular immune response may represent a physiological mechanism that enables FAW larvae to counteract the harmful effects of the insecticide, thus maintaining homeostasis. Moreover, the implications of this immune response on the efficacy of chlorantraniliprole when combined with bioinsecticides, or its interaction with other pathogens affecting FAW larvae, remains to be further investigated.

## 5. Conclusions

In this study, exposure of FAW larvae to chlorantraniliprole was shown to change the morphology of some hemocytes and influence the total number of hemocytes. Additionally, chlorantraniliprole enhanced cellular immunity in FAW larvae by stimulating the spread of plasmatocytes and increasing the areas of the cytoskeletons in hemocytes. This research serves as a foundational exploration into the cellular immune response in FAW larvae and provides a reference for investigating the immunological effects of insecticides on FAW larvae. 

## Figures and Tables

**Figure 1 insects-15-00586-f001:**
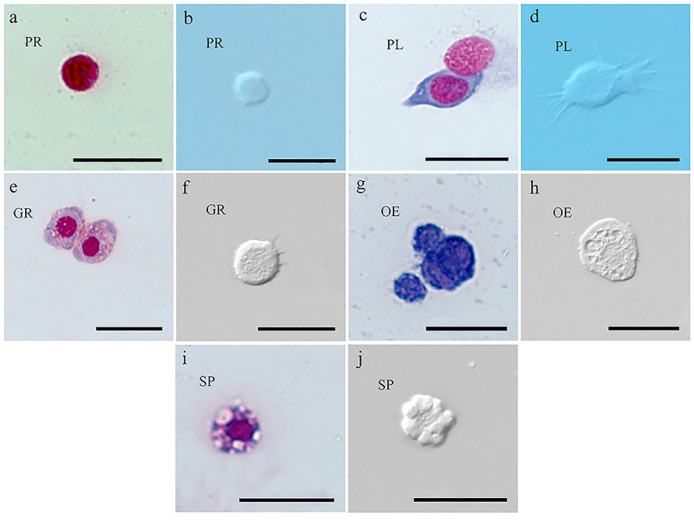
Types of hemocytes in *Spodoptera frugiperda* larvae. (**a**) The optical microscopy image of prohemocytes (PR). (**b**) The DIC of prohemocytes (PR). (**c**) The optical microscopy images of plasmatocytes (PL). (**d**) The DIC of plasmatocytes (PL). (**e**) The optical microscopy image of granulocytes (GR). (**f**) The DIC of granulocytes (GR). (**g**) The optical microscopy image of oenocytoids (OE). (**h**) The DIC of oenocytoids (OE). (**i**) The optical microscopy image of spherulocytes (SP). (**j**) The DIC of spherulocytes (SP). Scale bar: 20 μm.

**Figure 2 insects-15-00586-f002:**
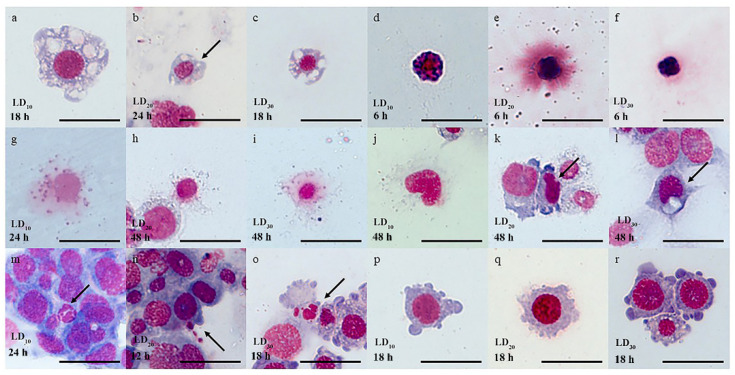
Effect of chlorantraniliprole on the hemocyte morphology in *Spodoptera frugiperda* larvae. (**a**–**c**) Cytoplasmic vacuolization, (**d**–**f**) cell pyknosis, (**g**–**i**) granulocyte degranulation, (**j**–**l**) nucleus deformation, (**m**–**o**) nucleus apoptosis, and (**p**–**r**) cell membrane deformation. Scale bar: 20 μm. The arrow in the figure points to the target hemocyte.

**Figure 3 insects-15-00586-f003:**
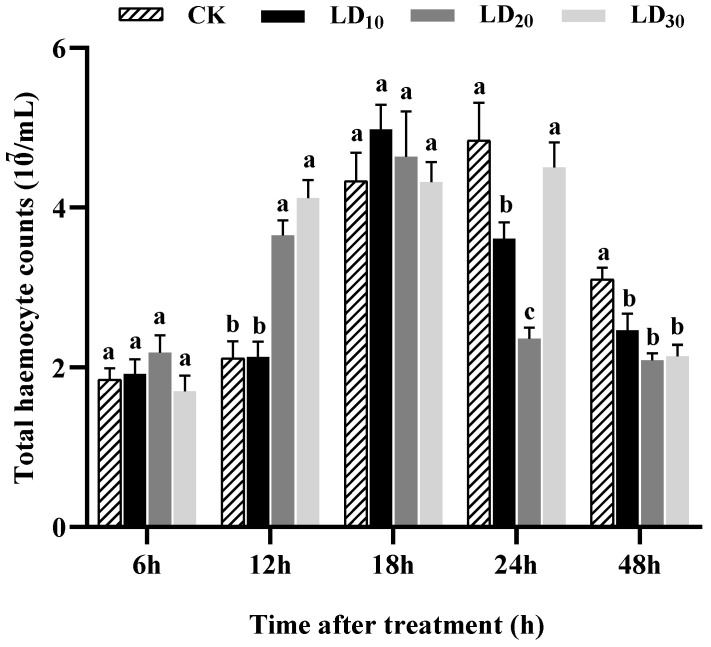
Effect of chlorantraniliprole on the total hemocyte counts in *Spodoptera frugiperda* larvae. The data in this figure are presented as means ± SD. Different letters on the bars represent significant differences (6, 12, and 18 h by one-way analysis of variance; 24 and 48 h by Kruskal–Wallis nonparametric tests, *p* < 0.05; n = 10, *df* = 3 and 36). The LD_10_, LD_20_, and LD_30_ groups were compared to the control group (CK) at the same time point.

**Figure 4 insects-15-00586-f004:**
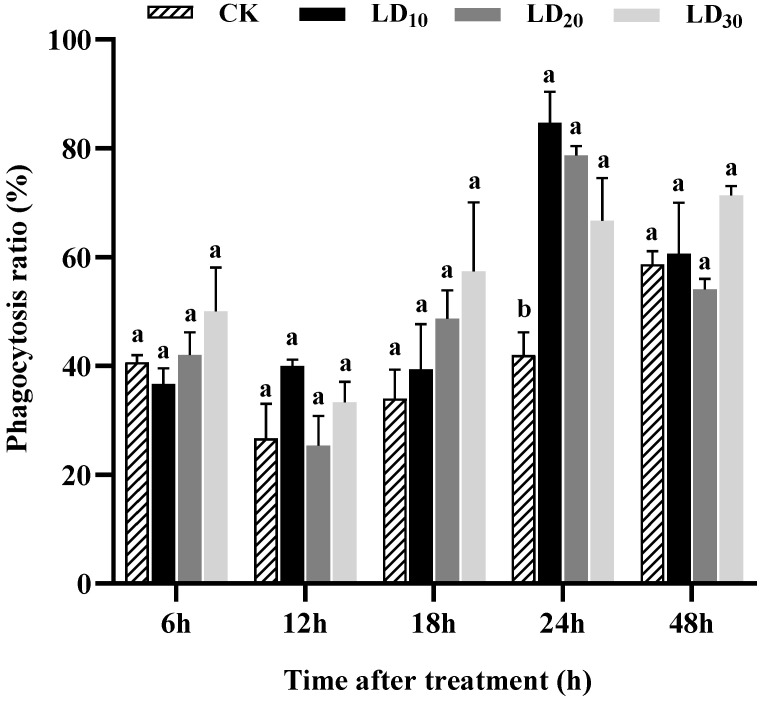
Effect of chlorantraniliprole on the phagocytosis ratio in *Spodoptera frugiperda* larvae. The data in this figure are presented as means ± SD. Different letters on the bars represent significant differences (12, 18, and 24 h by one-way analysis of variance; 6 and 48 h by Kruskal–Wallis nonparametric tests, *p* < 0.05; n = 3, *df* = 3 and 8). The LD_10_, LD_20_, and LD_30_ groups were compared to the control group (CK) at the same time point.

**Figure 5 insects-15-00586-f005:**
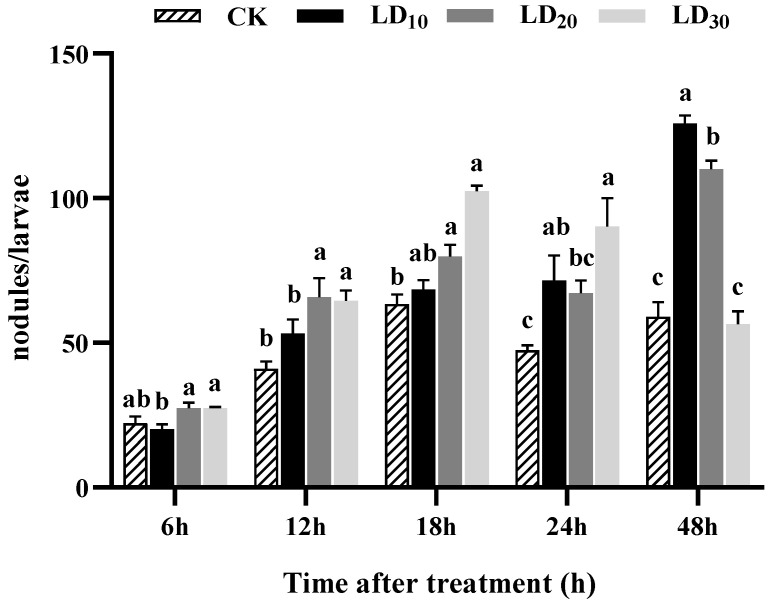
Effect of chlorantraniliprole on the nodulation in *Spodoptera frugiperda* larvae. The data in this figure are presented as means ± SD. Different letters on the bars represent significant differences (by one-way analysis of variance, *p* < 0.05; n = 5, *df* = 3 and 16). The LD_10_, LD_20_, and LD_30_ groups were compared to the control group (CK) at the same time point.

**Figure 6 insects-15-00586-f006:**
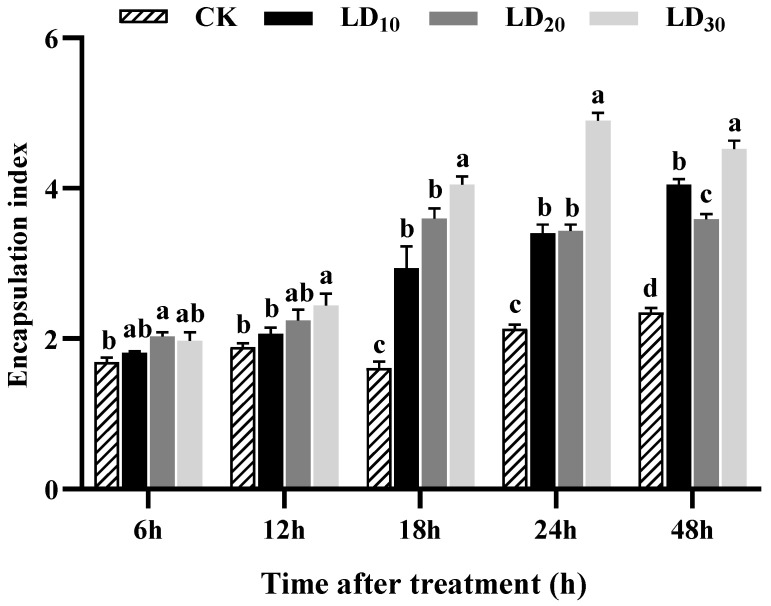
Effect of chlorantraniliprole on the encapsulation index in *Spodoptera frugiperda* larvae. The data in this figure are presented as means ± SD. Different letters on the bars represent significant differences (12, 24, and 48 h by one-way analysis of variance; 6 and 18 h by Kruskal–Wallis nonparametric tests, *p* < 0.05; n = 5, *df* = 3 and 16). The LD_10_, LD_20_, and LD_30_ groups were compared to the control group (CK) at the same time point.

**Figure 7 insects-15-00586-f007:**
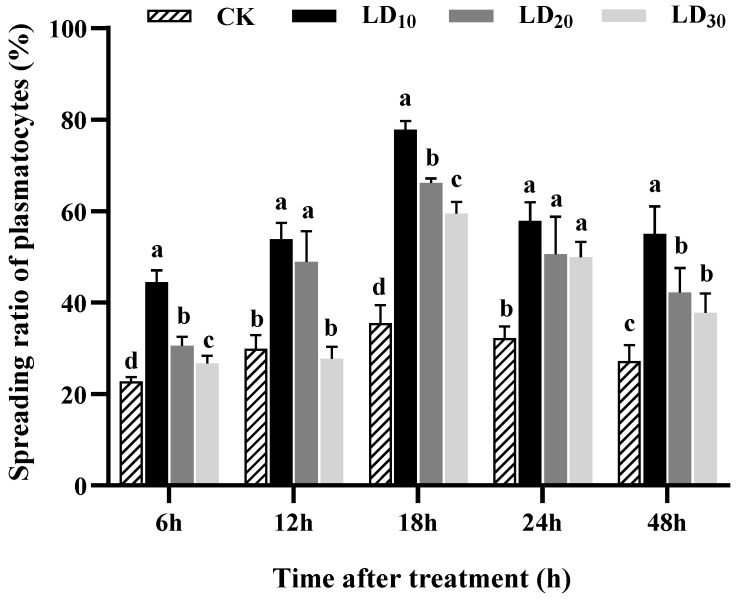
Effect of chlorantraniliprole on the plasmatocytes’ spreading ratio in *Spodoptera frugiperda* larvae. The data in this figure are presented as means ± SD. Different letters on the bars represent significant differences (by one-way analysis of variance, *p* < 0.05; n = 3, *df* = 3 and 8). The LD_10_, LD_20_, and LD_30_ groups were compared to the control group (CK) at the same time point.

**Figure 8 insects-15-00586-f008:**
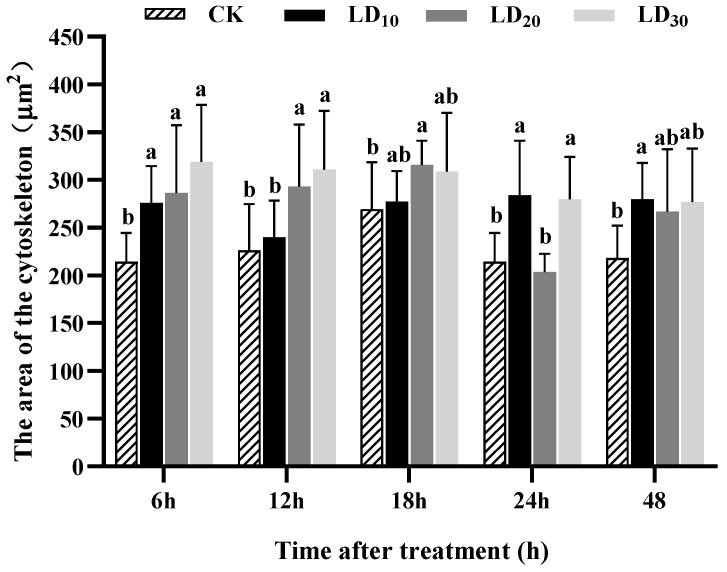
Effect of chlorantraniliprole on the area of the cytoskeleton in hemocytes of *Spodoptera frugiperda* larvae. The data in this figure are presented as means ± SD. Different letters on the bars represent significant differences (6 and 12 h by one-way analysis of variance; 18, 24, and 48 h by Kruskal–Wallis nonparametric tests, *p* < 0.05; n = 10, *df* = 3 and 36). The LD_10_, LD_20_, and LD_30_ groups were compared to the control group (CK) at the same time point.

**Figure 9 insects-15-00586-f009:**
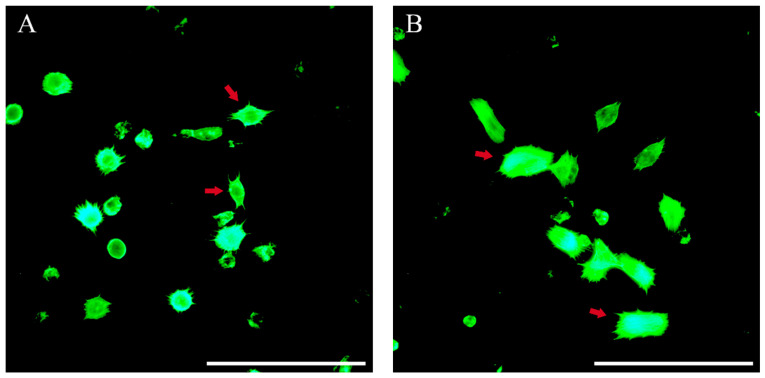
Fluorescent images of cytoskeletons in hemocytes of *Spodoptera frugiperda* larvae. (**A**) Control group: cytoskeletons in hemocytes treated with a 0.1% Triton X-100 aqueous solution for 48 h. (**B**) LD_30_ group: cytoskeletons in hemocytes treated with LD_30_ chlorantraniliprole for 48 h. Scale bar: 100 μm.

**Table 1 insects-15-00586-t001:** Toxicity of chlorantraniliprole to 4th instar larvae of *Spodoptera frugiperda*.

Regression Equation	LD_10_ (μg/g) (95% CL)	LD_20_ (μg/g) (95% CL)	LD_30_ (μg/g) (95% CL)	LD_50_ (μg/g) (95% CL)	χ^2^ (*df*)	*p*
y = −0.832 + 1.906x	0.581 (0.306~0.834)	0.988 (0.648~1.330)	1.449 (1.051~1.971)	2.731 (2.004~4.408)	5.963 (13)	0.948

LD, CL, χ^2^, *df*, and *p* indicate lethal dose, 95% confidence limits, chi-square value, degrees of freedom, and significance in Pearson’s chi-squared test, respectively. The *p*-value greater than the 0.05 level indicates that the observed data were close to the expected value of the model.

## Data Availability

The data presented in this study are available upon request from the corresponding author.

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
