# Peer review of "Chlorantraniliprole Enhances Cellular Immunity in Larvae of Spodoptera frugiperda (Smith) (Lepidoptera: Noctuidae)"

_insects, 2024, doi:10.3390/insects15080586_

Round 1

Reviewer 1 Report (New Reviewer)

Comments and Suggestions for Authors

Comments to the Author

Investigating the effects of chemical pesticides on the immune system of agricultural insect pests provides important insights into their cellular immune responses to insecticide exposure. In this manuscript, Liu et al. report an enhanced cellular immune response in larvae of Spodoptera frugiperda following exposure to the insecticide chlorantraniliprole. The study, which makes a valuable contribution to the understanding of lepidopteran immunity, is methodologically well structured, the analysis is quite thorough, and the conclusions are sufficiently supported by the data obtained. Overall, the manuscript is well written, and I recommend its publication, subject to minor revisions, which I report below:

Entire manuscript. Update taxonomic and author status of each species when it is mentioned for the first time. For example: Conogethes punctiferalis (Guenée) (Lepidoptera: Crambidae).

Figures and tables. In the captions, the name of the species should be given in full  and not as acronym.

Results. I recommend presenting statistical values​​ only in image captions and not in the main text to improve readability. In the text, I suggest reporting only the P value, while comprehensive statistical details should be limited to the image captions. However, compliance with this suggestion depends on the journal's specific guidelines. Likewise, the sample sizes (n) should be included in the charts, positioned either above or below the bars or detailed in the figure captions.

Suggestion for Table S1 caption. Replace “List of artificial diet of FAW larvae” with “Composition of artificial diet for rearing Spodoptera frugiperda larvae”.

Line 44: Insert a citation after “diverse array of pathogens”.

Line 46:  Insert a citation after “invasion”.

Line 101. The characterization of the insect immune system as "robust" should be approached with caution. Instead, a more accurate description would be: "Insects exhibit complex and multifaceted immune systems, which provide protection against a wide range of pathogens, contributing to their extensive distribution across diverse ecosystems".

Line 111. The term "control agent" in relation to an insecticide seems inadequate. I would suggest: “Chlorantraniliprole, a systemic insecticide with a broad spectrum of activity against Lepidoptera pests, is classified as an essential emergency measure for the control of FAW in the context of integrated pest management (IPM)”.  

Lines 129-130. The objectives of the study need to be clarified. In particular, the authors investigate the effects of chlorantraniliprole exposure on various aspects such as morphology and frequency of ??? These studies focus primarily on hemocytes, as indicated in the previous sentence (lines 128-129). Improving the consistency and coherence of the text is crucial to improving its overall fluency.

Line 175. Specify sterilization time in ethanol 75%.

Lines 277-278. The results are already shown in Table 1 and should not be repeated twice. Review according to journal guidelines.

Line 439. Add a citation after “Insect cellular immunity is primarily mediated by hemocytes”.

Line 452. Add a citation after “Pesticide exposure can adversely affect the morphology of hemocytes in insects”.

Comments on the Quality of English Language

The quality of the English language is quite good. It can still be improved, but it is not a defect. Overall, the authors of this work have written a good manuscript.

Author Response

Dear Reviewer,

Thank you for your thorough and constructive feedback on the manuscript titled “Chlorantraniliprole enhances cellular immunity in larvae of Spodoptera frugiperda (Smith) (Lepidoptera: Noctuidae)” with the reference number insects-3095374.

Your positive remarks on the manuscript's quality of writing are greatly appreciated and serve as a significant encouragement to our research. Your endorsement for publication, subject to minor revisions, has been taken to heart. We have meticulously attended to each of your suggestions and have made the necessary amendments to the manuscript. Furthermore, to enhance the clarity and precision of our work, we have engaged the services of a professional language editor. With the utmost diligence, we have incorporated your feedback to ensure that the revised manuscript meets the high standards of the journal. Below, we have addressed the specific review comments that required further elaboration, which are highlighted in blue.

We are deeply appreciative of the time and effort you have invested in evaluating our manuscript.

Specific Comments:

Entire manuscript. Update taxonomic and author status of each species when it is mentioned for the first time. For example: Conogethes punctiferalis (Guenée) (Lepidoptera: Crambidae).

Response: We extend our heartfelt gratitude for the insightful comments and constructive suggestions you have offered. Your expertise and guidance are instrumental in enhancing the quality of our work. In response to your request and in alignment with the journal's guidelines, we have meticulously revised the manuscript to ensure both clarity and strict adherence to taxonomic standards. Specifically, we have now incorporated the author status for each species upon its initial mention in the text. This amendment addresses the need for academic precision while introducing the species to our readers. We acknowledge the importance of taxonomic detail. However, we also recognize the value of a readable and accessible manuscript. Thus, after the first citation, we have omitted the inclusion of the taxonomic status for each species. This decision was carefully considered and aimed at preserving the manuscript's conciseness and readability. For example: Conogethes punctiferalis (Guenée).

Figures and tables. In the captions, the name of the species should be given in full and not as acronym.

Response: Thank you for your valuable feedback on our manuscript. We appreciate the attention to detail you have provided, particularly regarding the presentation of species names in the captions of figures and tables. In response to your suggestion, we have revised the manuscript to ensure that the full names of all species are now used in the captions, rather than acronyms. This change has been applied consistently throughout the document to maintain clarity.

Results. I recommend presenting statistical values​​ only in image captions and not in the main text to improve readability. In the text, I suggest reporting only the P value, while comprehensive statistical details should be limited to the image captions. However, compliance with this suggestion depends on the journal's specific guidelines. Likewise, the sample sizes (n) should be included in the charts, positioned either above or below the bars or detailed in the figure captions.

Response: We greatly appreciate your thoughtful suggestions and have taken your advice to enhance the clarity and readability of our manuscript. In accordance with your recommendation, we have relocated the detailed statistical values, including the sample sizes (n), and degrees of freedom, to the image captions. This relocation ensures that the main text remains uncluttered and focused on the interpretation of the results. We have also included the P values and F values in the main text, as you suggested, to succinctly convey the significance of our findings.

Suggestion for Table S1 caption. Replace “List of artificial diet of FAW larvae” with “Composition of artificial diet for rearing Spodoptera frugiperda larvae”.

Response: Thank you for your detailed and constructive feedback on our manuscript. We have taken your suggestion regarding the caption for Table S1 into careful consideration. We have revised the caption to read: “Composition of artificial diet for rearing Spodoptera frugiperda larvae,” replacing the previous caption “List of artificial diet of FAW larvae.”

Line 44: Insert a citation after “diverse array of pathogens”.

Response: Thank you for your thorough review and for pointing out the need for additional citation at line 44 of our manuscript. We have carefully considered your suggestion and have inserted a citation at line 44.

[1] Silva, R.C.M.C.; Ramos, I.B.; Travassos, L.H.; Mendez, A.P.G.; Gomes, F.M. Evolution of Innate Immunity: Lessons from Mammalian Models Shaping Our Current View of Insect Immunity. J Comp Physiol B 2024, 194, 105–119, doi:10.1007/s00360-024-01549-1.

Line 46: Insert a citation after “invasion”.

Response: Thank you for your meticulous review. We have taken your suggestion to heart and have now included a citation at line 46.

[2] Andoh, M.; Ueno, T.; Kawasaki, K. Tissue-Dependent Induction of Antimicrobial Peptide Genes after Body Wall Injury in House Fly (Musca domestica) Larvae. DD&T 2018, 12, 355–362, doi:10.5582/ddt.2018.01063.

Line 101. The characterization of the insect immune system as "robust" should be approached with caution. Instead, a more accurate description would be: "Insects exhibit complex and multifaceted immune systems, which provide protection against a wide range of pathogens, contributing to their extensive distribution across diverse ecosystems".

Response: We greatly appreciate your insightful comments on our manuscript, particularly the nuanced perspective you have offered on the characterization of the insect immune system at line 95. In light of your guidance, we have revised the sentence.

Line 111. The term "control agent" in relation to an insecticide seems inadequate. I would suggest: “Chlorantraniliprole, a systemic insecticide with a broad spectrum of activity against Lepidoptera pests, is classified as an essential emergency measure for the control of FAW in the context of integrated pest management (IPM)”.

Response: Thank you for your detailed review and for the specific feedback on the use of the term “control agent” at line 106 of our manuscript. We have revised the sentence.

Lines 129-130. The objectives of the study need to be clarified. In particular, the authors investigate the effects of chlorantraniliprole exposure on various aspects such as morphology and frequency of ??? These studies focus primarily on hemocytes, as indicated in the previous sentence (lines 128-129). Improving the consistency and coherence of the text is crucial to improving its overall fluency.

Response: We thank you for your careful reading and for pointing out the need for clarification in the objectives of our study at lines 124-125. We have clarified the objectives to more explicitly state the scope of our investigation. The revised sentence now reads: “We examined the impacts of chlorantraniliprole exposure on morphology and abundance of hemocytes, as well as phagocytosis, encapsulation, and nodulation in FAW larvae.”

Line 175. Specify sterilization time in ethanol 75%.

Response: We appreciate your attention to the methodological details in our manuscript. We would like to express our sincere apologies for the oversight in our manuscript regarding the description of the sterilization process using 75% ethanol on the abdominal prolegs of the larvae. To address this, we have revised the manuscript to include a clear description of the sterilization step. The revised text now specifies at line 160: “The abdominal prolegs were then sterilized by swabbing three times with 75% ethanol and subsequently punctured using an insect needle..”

Lines 277-278. The results are already shown in Table 1 and should not be repeated twice. Review according to journal guidelines.

Response: Thank you for your insightful comments and for drawing our attention to the potential redundancy in lines 277-278 of our manuscript. I understand your concern regarding the repetition of results in the manuscript. In accordance with the journal's guidelines, I have revised the manuscript to ensure that the results presented in Table 1 are not duplicated elsewhere in the text. The revised text now specifies at line 274: “The chlorantraniliprole toxicity results for the 4th instar larvae of FAW are shown in Table 1. The results of the bioassay demonstrated the high insecticidal efficacy of chlorantraniliprole against 4th instar larvae of the FAW.”

Line 439. Add a citation after “Insect cellular immunity is primarily mediated by hemocytes”.

Response: Thank you for your valuable feedback and for highlighting the need for a citation at line 439 of our manuscript. In response to your recommendation, we have added a citation to the sentence at line 434.

[16] Strand, M.R. Insect Hemocytes and Their Role in Immunity. Insect Biochem. Molec. 2002, 32, 1295–1309, doi:10.1016/S0965-1748(02)00092-9.

Line 452. Add a citation after “Pesticide exposure can adversely affect the morphology of hemocytes in insects”.

Response: We appreciate the opportunity to enhance the credibility and depth of our manuscript through proper citation, as suggested by your insightful feedback at line 448. We have added a citation to the sentence.

[62] Perveen, N.; Ahmad, M. Toxicity of Some Insecticides to the Haemocytes of Giant Honeybee, Apis dorsata F. under Laboratory Conditions. Saudi J. Biol. Sci. 2017, 24, 1016–1022, doi: 10.1016/j.sjbs.2016.12.011.

[63] Huang, Q.; Zhang, L.; Yang, C.; Yun, X.; He, Y. The Competence of Hemocyte Immunity in the Armyworm Mythimna separata Larvae to Sublethal Hexaflumuron Exposure. Pesti. Biochem. Phys. 2016, 130, 31–38, doi: 10.1016/j.pestbp.2015.12.003.

Thank you once again for your constructive feedback and for the opportunity to refine our manuscript. I hope these revisions meet your expectations and the requirements of the journal.

Reviewer 2 Report (New Reviewer)

Comments and Suggestions for Authors

In the present manuscript authors investigated the effect of sublethal doses of chlorantraniliprole on the cellular immune response of the lepidopteran insect Spodoptera frugiperda. The results are interesting and suggest an increased immune response via Ca2+ flux alteration in hemocytes. The manuscript requires very few minor changes for publication, which are included as comments in the pdf attached.

Author Response

Dear Reviewer,

Thank you for your thorough and constructive feedback on the manuscript titled “Chlorantraniliprole enhances cellular immunity in larvae of Spodoptera frugiperda (Smith) (Lepidoptera: Noctuidae)” with the reference number insects-3095374.

Your insightful comments and positive remarks on our work are greatly valued. We are pleased that you thought the results of our investigation interesting, particularly regarding the potential for an increased immune response through the alteration of Ca2+ flux in hemocytes. Your feedback is instrumental in enhancing the quality of our research and its presentation. We have carefully considered the revisions you suggested, and have made the appropriate adjustments to our manuscript. The changes have been meticulously implemented. Below, we have addressed the specific review comments that required further elaboration.

We are deeply appreciative of the time and effort you have invested in evaluating our manuscript.

Specific Comments:

Line 36. this is a repetition of what already said above. Indeed these are the results. Here authors should explain what their results SUGGEST, with a bit of speculation of why they observed this increase in immune response at sublethal doses.

Response: We extend our heartfelt gratitude for your valuable feedback and the opportunity to revisit our manuscript. Your observation regarding the repetition in the abstract has been noted, and we have taken it into consideration in our revisions. In response to your suggestion for including speculative elements in the abstract, we have given it due thought. However, we have chosen to maintain the abstract's focus on the direct findings of our study. We believe that this sentence in question encapsulates the core conclusion of our research. We aim to provide a clear and precise summary of our empirical results without delving into speculative interpretations that are not directly supported by our data. While speculation can be a valuable component of scientific discourse, we believe that it is essential to distinguish between empirical evidence and theoretical possibilities. By keeping the abstract centered on our results, we can avoid any potential misinterpretation. Additionally, we feel that speculative elements are more appropriately addressed in the discussion section. Here, we speculate on why an increase in immune response was observed at sublethal doses at lines 536-538: “This heightened cellular immune response may represent a physiological mechanism that enables FAW larvae to counteract the harmful effects of the insecticide, thus maintaining homeostasis.” At the same time, we have refined the sentence to enhance its brevity and eliminate any redundancy. The revised sentence now reads at lines 35-37: “These results suggest that exposure to chlorantraniliprole results in an enhanced immune response function in FAW larvae, which may be mediated by cytoskeletal changes and plasmatocyte spreading.”

Line 94. Please consider also:

- Annoscia, D., Di Prisco, G., Becchimanzi, A., Caprio, E., Frizzera, D., Linguadoca, A., et al. (2020). Neonicotinoid Clothianidin reduces honey bee immune response and contributes to Varroa mite proliferation. Nature Communications 11, 5887. doi: 10.1038/s41467-020-19715-8.

- Di Prisco G, Cavaliere V, Annoscia D, Varricchio P, Caprio E, Nazzi F, Gargiulo G, Pennacchio F. Neonicotinoid clothianidin adversely affects insect immunity and promotes replication of a viral pathogen in honey bees. Proc Natl Acad Sci U S A. 2013 Nov 12;110(46):18466-71. doi: 10.1073/pnas.1314923110. Epub 2013 Oct 21. PMID: 24145453; PMCID: PMC3831983.

Response: Thank you for drawing our attention to the important studies by Annoscia et al. (2020) and Di Prisco et al. (2013). We acknowledge the significance of these works in the context of our research. We have now expanded our literature review to include a discussion of the findings of Annoscia et al. (2020) and Di Prisco et al. (2013) in the introduction (Lines 85-87). The added sentence is: “Similarly, the neonicotinoid clothianidin had a negative impact on immune responses, which can boost the proliferation of honey bee parasites and pathogens [27,28]”

[27] Annoscia, D.; Di Prisco, G.; Becchimanzi, A.; Caprio, E.; Frizzera, D.; Linguadoca, A.; Nazzi, F.; Pennacchio, F. Neonicotinoid Clothianidin Reduces Honey Bee Immune Response and Contributes to Varroa Mite Proliferation. Nat Commun 2020, 11, 5887, doi:10.1038/s41467-020-19715-8.

[28] Di Prisco, G.; Cavaliere, V.; Annoscia, D.; Varricchio, P.; Caprio, E.; Nazzi, F.; Gargiulo, G.; Pennacchio, F. Neonicotinoid Clothianidin Adversely Affects Insect Immunity and Promotes Replication of a Viral Pathogen in Honey Bees. Proc. Natl. Acad. Sci. U.S.A. 2013, 110, 18466–18471, doi:10.1073/pnas.1314923110.

Line 99. I would say: "organophosphorus insecticides enhanced." to avoid confounding the reader.

Response: We sincerely apologize for any confusion caused by the ambiguous use of “it” in our previous submission. We appreciate your attention to detail and have taken your feedback to heart. To eliminate any ambiguity and to improve the clarity of our manuscript, we have revised the sentence as follows at line 91: “Additionally, when G. mellonella larvae were infected with Bacillus thuringiensis Berliner, their cellular and humoral immune responses were enhanced, enabling them to resist its effects.”

Line 101. This is a repetition of what is already said above. It can be deleted.

Response: We are grateful for your meticulous review and your efforts to ensure our manuscript's conciseness and coherence. Your observation regarding the redundancy of the sentence in question has been duly noted. After thoughtful deliberation, we have chosen to retain the sentence, as we believe it serves a pivotal role in our narrative. Specifically, it provides a smooth transition to the subsequent discussion on Spodoptera frugiperda, a widespread pest. However, we have taken your feedback to heart and have revised the sentence to enhance its distinctiveness and relevance. The revised sentence now reads at line 94: “Insects exhibit complex and multifaceted immune systems, which provide protection against a wide range of pathogens, contributing to their extensive distribution across diverse ecosystems.”

Line 434. I would not say that chemical agents cause infections. This part should be deleted.

Response: We appreciate your careful reading of our manuscript and your insightful comment regarding the statement about chemical agents causing infections. Your guidance is invaluable in helping us present the most accurate information. Upon reflection, we understand and acknowledge your concern that the phrase might be misleading. We agree that the term "infections" is typically associated with biological pathogens rather than chemical agents. To clarify our statement and avoid any misconceptions, we have decided to revise the section for greater precision. The revised text now reads at line 430: “Innate immunity serves as the primary initiator of the immune response against pathogenic invaders, forming the initial line of defense in the host.”

Line 471. I would change this part in "the hemocyte count of insect depends by the different molecules, doses and species considered."

Response: We thank you for your attentive review and the specific suggestion to refine the statement regarding the hemocyte count in insects. Your guidance is greatly appreciated and has prompted us to clarify this aspect of our manuscript. We concur that the original phrasing may have been ambiguous and could be improved for precision and readability. The revised sentence now reads at line 467: “The hemocyte count in insects is influenced by a variety of factors, including the specific molecules involved, the doses administered, and the particular species considered.”

Line 504. It seems that in nodulation and spreading at 48h the response follow a two phased dose-response relationship (hormesis). Hormesis is when low-dose amounts of a compound have a beneficial effect, while high-dose are inhibitory. Please consider.

Response: We sincerely appreciate your insightful suggestion regarding the potential hormetic dose-response relationship in our study. Your expertise in identifying possible hormetic effects is highly valued. After carefully reconsidering our data on nodulation and spreading at 48 hours, we have conducted a thorough statistical analysis to evaluate the presence of a hormetic effect. While we observed that low doses of chlorantraniliprole significantly increased the number of nodules compared to the control group, the reduction in nodule number at high doses did not reach statistical significance. This finding suggests that, although there is a trend towards a biphasic response, the hormetic effect may not be fully expressed in our results. Furthermore, in the plasmatocyte-spreading assay, we found that chlorantraniliprole at all doses tested significantly increased the spreading rate compared to the control group. These findings are inconsistent with the characteristics of a hormetic response. Given these considerations, and in the interest of maintaining the scientific rigor and clarity of our discussion, we have chosen not to elaborate on the hormetic effect in this section of the manuscript.

Line 532. this speculation is interesting and can be used in the abstract, instead of repeating the finding of the present work.

Response: We are sincerely grateful for your innovative suggestion to include the engaging speculation on the role of Ca2+ in cellular immunity within our abstract. Your perspective offers a compelling and thought-provoking angle that could potentially enhance the narrative of our research. Upon thorough deliberation, we have chosen to keep the abstract centered on the empirical findings of our study. Although the implications of Ca2+ flux on hemocyte behavior are both captivating and pertinent to the wider scientific discourse, these insights are based on a review of the literature rather than the direct outcomes of our experiments. Our objective is to provide a transparent and succinct summary of our experimental results, ensuring that the abstract serves as an accurate and direct representation of our work. We believe that discussions of the broader implications, such as the potential role of calcium ions, are more appropriately placed within the main text of the paper. There, they can be examined in depth, contextualized by our data, and aligned with the existing body of literature. Thank you again for your constructive feedback, which has significantly contributed to our reflection on the most effective way to convey our research. To avoid redundancy in the abstract, we have revised the sentence as follows at line 36: “These results suggest that exposure to chlorantraniliprole results in an enhanced immune response function in FAW larvae, which may be mediated by cytoskeletal changes and plasmatocyte spreading.”

Line 550. I would say: "artificial diet components".

Response: We appreciate your suggestion to refine our terminology in the manuscript. Your expertise and attention to detail are highly valued in enhancing the clarity of our work. In response to your recommendation, we have revised the term to better align with the specific context of our study. The revised sentence now reads: “Composition of artificial diet for rearing Spodoptera frugiperda larvae.”

Thank you once again for your constructive feedback and for the opportunity to refine our manuscript. I hope these revisions meet your expectations and the requirements of the journal.

This manuscript is a resubmission of an earlier submission. The following is a list of the peer review reports and author responses from that submission.

Round 1

Reviewer 1 Report

Comments and Suggestions for Authors

The manuscript entitled " Sublethal chlorantraniliprole exposure enhances cellular immunity in larvae of Spodoptera frugiperda (Smith) (Lepidoptera:Noctuidae)" describes the impact of a commonly used insecticide for the management of Spodoptera frugiperda on different aspects of the cellular immunity of this key pest. In particular, authors investigated the potential immunotoxic effects of chlorantraniliprole on the abundance, morphology and function of hemocytes, as well as on some cellular process mediated by these cells, e.g., encapsulation, nodulation and phagocytosis.

The data presented in this study are novel, and the research represents one of the first investigations into the immunotoxic impact of insecticides on such a significant pest. However, substantial improvements are necessary throughout the manuscript before it can be considered for potential publication in the journal. My primary recommendation is to improve the language used in the manuscript and the data analysis and results section. Additionally, it is crucial to provide a more thorough discussion of the potential scenarios in which these findings could occur.

General comments

-        The language of the manuscript needs to be highly improved with the help of a native speaker. The reading is often hard, particularly in the results section.

-        Material and methods sections seems to be well described, although some essential information needs to be provided by the authors, such as references for the used formulas and statistical methods used in the data analysis. Specifically, there appears to be inconsistency between the data analysis methods (ANOVA) and the results presented in the graphs (comparisons between two groups rather than four groups). Additionally, information regarding the homogeneity and normality of the dataset before ANOVA, as well as post-hoc test results, is lacking.

-        The results section is sometimes redundant with the materials and methods section. Authors should improve the quality of the provided figures (in particular the figure 1 and 2. Also, figure 4 reports the tested doses as LC, while in all the manuscript authors reported LD). Most importantly, authors should elucidate the results by first indicating what is statistically significant, followed by explaining various trends, even if they are not statistically significant, to prevent misleading the reader during interpretation.

-        The discussions are adequate, but it is essential to better contextualize in which scenarios these findings could occur and be beneficial (e.g., chlorantraniliprole enhances cellular immunity of FAW and could increase pest resistance to pathogens. It is expected this could occurs in the field and against what pathogens could FAW be more resistant? Indeed, the pest could be more resistant against pathogens naturally living in the (agro)ecosystem and/or microbials used as biopesticides for pest control. In both cases there would be alterations on the insect-pathogen interactions and/or on the IPM strategies).  

Specific comments:

L143-145: authors should report the reference of the study where the chlorantraniliprole sublethal concentrations used in this study were estimated.

L146: authors should provide references for the formula used to calculate the total hemocyte count, as well as for the phagocytosis (L199-201) and encapsulation (221-223)

L248-251: Please, specify if the data used for the ANOVA were homogenous and normally distributed and the post-hoc test used after the ANOVA. Indeed, looking at the figure 1-8 it seems that authors compared differences between two groups (control and each tested concentration, e.g., control-LD10; control-LD20; control-LD30) instead of showing differences among three or more groups (which is the main objective of the ANOVA). Authors should provide clarification of these issues providing the needed information). As a suggestion, authors could employ a GLM to examine the impact of each independent variable (e.g., pesticide dose, time after treatment, and their interaction) to elucidate which variable influences the tested parameters. This approach would facilitate a clearer understanding of the results and enable more insightful discussions.

L254-255: this part belongs to the material and method section.

L294: authors reported only one degree of freedom of ANOVA output, that is that one referring to the number of groups considered in the analysis. However, it is needed to specify also the degree of freedom indicating the number of observations, which is needed to understand the sample size tested for each parameter. Please, provide the missing information. See also the degree of freedom reported in section 3.4, 3.5, 3.6, 3.7 and 3.8.

L277-286: it is not clear if the authors assessed the impact of sublethal concentrations of chlorantraniliprole on hemocyte morphology for all the immune cells (i.e., prohemocytes, plamastocytes, granulocytes, oenocytoids, spherulocytes and cystocytes) and for each time after chemical exposure (i.e., 6, 12, 18, 24 and 48 hours). Indeed, authors reported results only for some hemocytes and some times after the exposure.

L292-300: this is inconsistent with the results showed in the graph of figure 3. Authors reported that “Except for LD10, in the treatment group, it was significantly higher than that in the control group at 12 h”. However, it is not clear “what was higher” and “higher than what”.

L296: Remove similarly since the LD30 at 48h was significantly higher than LD30 reported at 24h.

L298-299: The authors should clarify the results by first indicating what is statistically significant and then explaining various trends, even if not statistically significant, to avoid misleading the reader during interpretation.

L410-419: While it is explained that a morphological change in the hemocytes has been observed, it remains uncertain what consequences this may have on physiology and/or immune defense of FAW, especially regarding which hemocytes were affected. Authors could improve their discussion by providing more information on this topic.

L420-431: The alteration of THC serves as a crucial starting point for studying the immunotoxic effects of insecticides. However, it is also necessary to explore the potential consequences of insecticide exposure on the different hemocyte classes (DHC) to draw stronger conclusions regarding the potential effects on cellular defense processes driven by various classes of insecticides, such as phagocytosis, encapsulation, and nodulation, which were investigated in this study. Authors could consider to incorporate these aspects to their discussion.

L472-475: The authors should further discuss in which scenario chlorantraniliprole, by influencing the immunity of FAW, may alter susceptibility to pathogens. For instance, they could explore pathogens commonly found in the field, as well as those used as bio-insecticides, which could potentially modify the insect-pathogens interaction in nature and the IPM programs using different control strategie (e.g., chemical and bio-insecticides).

Comments on the Quality of English Language

-        The language of the manuscript needs to be highly improved with the help of a native speaker. The reading is often hard, particularly in the results section.

Author Response

Dear Reviewers,

Thanks for providing us with this great opportunity to submit a revised version of our manuscript entitled "Sublethal chlorantraniliprole exposure enhances cellular immunity in larvae of Spodoptera frugiperda (Smith) (Lepidoptera: Noctuidae)", with the reference number insects-2958906.

We have studied the comments carefully and made major rectifications which we hope meet with approval. And we have made significant improvements to the language used in the manuscript. Below are our detailed responses to the reviewer's comments, highlighted in blue.

1. L143-145: authors should report the reference of the study where the chlorantraniliprole sublethal concentrations used in this study were estimated.

Response: We sincerely appreciate the valuable comments. As suggested by the reviewer, we have added the Larval Toxicity Bioassay in 2.2 of Materials and Methods (Lines 148-159). We have also added the result of Larval Toxicity Bioassay in 3.1 of Results (p.6., Lines 258-264, Table 1).

2. L146: authors should provide references for the formula used to calculate the total hemocyte count, as well as for the phagocytosis (L199-201) and encapsulation (221-223).

Response: Thank you for the detailed review. We have checked the literature carefully and added the references for the total hemocyte count (p.4., Lines 188), as well as for the phagocytosis (p.5., Lines 201) and encapsulation (p.5., Lines 221).

The references are as follows:

Li, T.; Liu, H.; Wang, G; Li, Y; Yu, H.; Yan, D.; Guo, Y.; Zhang, T.; Chen, P. Reasons for Changes of Hemocyte Densities and the Relationship between Hemocyte Density and High Temperature Resistance of Bombyx Mori Larvae. Acta Entomol. Sin. 2022, 65, 130–143. doi: 10.16380/j.kcxb.2022.02.002.

Walkowiak-Nowicka, K.; Nowicki, G.; Kuczer, M.; Rosiński, G. New Activity of Yamamarin, an Insect Pentapeptide, on Immune System of Mealworm, Tenebrio Molitor. Bull. Entomol. Res. 2018, 108, 351–359. doi: 10.1017/S0007485317000839.

Hu, Q.; Wei, X.; Li, Y.; Wang, J.; Liu, X. Identification and Characterization of a Gene Involved in the Encapsulation Response of Helicoverpa Armigera Haemocytes. Insect Mol. Biol. 2017, 26, 752–762, doi: 10.1111/imb.12336.

3. L248-251: Please, specify if the data used for the ANOVA were homogenous and normally distributed and the post-hoc test used after the ANOVA. Indeed, looking at Figure 1-8 it seems that authors compared differences between two groups (control and each tested concentration, e.g., control-LD10; control-LD20; control-LD30) instead of showing differences among three or more groups (which is the main objective of the ANOVA). Authors should provide clarification of these issues providing the needed information). As a suggestion, authors could employ a GLM to examine the impact of each independent variable (e.g., pesticide dose, time after treatment, and their interaction) to elucidate which variable influences the tested parameters. This approach would facilitate a clearer understanding of the results and enable more insightful discussions.

Response: Thank you for your comments on our data analysis methodology.

(a) We used the Shapiro-Wilk and Levene's tests to test the assumptions of normality and homogeneity of variance before analysis. For the data to meet the assumptions of normality and homogeneity of variance, we have used one-way ANOVA of variance followed by LSD's multiple comparison analysis for the analysis of significant differences. For the normality assumption that was not met, the non-parametric Kruskal-Wallis test of variance followed by multiple comparison tests have been used for the analysis of significant differences (p.6., Lines 252-264).

(b) To show differences among three or more groups, we have used letters instead of "*" to indicate differences between groups in Figure 3-8.

(c) We have given due consideration to the reviewer's comments to utilize a GLM for our analysis. Our primary focus was on comparing the effects of various doses of chlorantraniliprole on the hemocytes and cellular immunity of the FAW at the same time. Nevertheless, we will take the reviewer's advice into account and incorporate the use of GLM in our future studies to better account for the effects of both dose and time on our experimental results.

Figure 3. Effect of chlorantraniliprole on the total hemocyte count in FAW larvae. Data in the figure are presented as mean ± SD. The different treatment groups are compared to the control group at the same time. Different letters on the bars represent significant differences determined by one-way ANOVA of variance and Kruskal–Wallis nonparametric tests (P < 0.05).

4. L254-255: this part belongs to the material and method section.

Response: Thank you for the rectification. We have removed this sentence “The hemocytes of the FAW larvae were stained with Giemsa-Wright's stain and then identified by the positive fluorescence microscope.” (p.6., Lines 274).

5. L294: authors reported only one degree of freedom of ANOVA output, that is that one referring to the number of groups considered in the analysis. However, it is needed to specify also the degree of freedom indicating the number of observations, which is needed to understand the sample size tested for each parameter. Please, provide the missing information. See also the degree of freedom reported in sections 3.4, 3.5, 3.6, 3.7, and 3.8.

Response: Thank you for your precious comments. We have now added the degrees of freedom for both between-group and within-group comparisons in sections 3.3 through 3.8. For instance, “at 12 h, the total number of hemocytes in the LD20 and LD30 groups was significantly higher than that in the control and LD10 group (df = 3, 36; F = 25.402; p < 0.0001)”. The notation 'df = 3, 36' indicates that the degrees of freedom are 3 for between-group comparisons and 36 for within-group comparisons, respectively."

6. L277-286: it is not clear if the authors assessed the impact of sublethal concentrations of chlorantraniliprole on hemocyte morphology for all the immune cells (i.e., prohemocytes, plasmatocytes, granulocytes, oenocytoids, spherulocytes and cystocytes) and for each time after chemical exposure (i.e., 6, 12, 18, 24 and 48 hours). Indeed, authors reported results only for some hemocytes and some times after the exposure.

Response: We sincerely appreciate the valuable comments. Sublethal chlorantraniliprole exposure severely affected the major morphology of hemocytes in our study, so it was impossible to determine exactly which type of hemocyte morphology was affected. Furthermore, our study examined the morphology of hemocytes at different times after sublethal chlorantraniliprole exposure. However, we found that there were no morphological changes in hemocytes at certain times, so we described the obvious changes at some times in the results (p.7., Lines 300-310).

7. L292-300: this is inconsistent with the results showed in the graph of figure 3. Authors reported that “Except for LD10, in the treatment group, it was significantly higher than that in the control group at 12 h”. However, it is not clear “what was higher” and “higher than what”.

Response: We apologize for our inaccurate description of the results, we have corrected it (p.8., Lines 317-321). The relevant content is as follows: “At 12 h, the total number of hemocytes in the LD20 and LD30 groups was significantly higher than that in the control and LD10 groups (df = 3, 36; F = 25.402; p < 0.0001). At 24 h, the number of hemocytes in the LD20 and LD20 groups was significantly lower than that of the control group (df = 3, 36; F = 23.701; p < 0.0001)”.

8. L296: Remove similarly since the LD30 at 48h was significantly higher than LD30 reported at 24h.

Response: Thank you for the suggestion. We have removed “similarly” (Lines 321).

9. L298-299: The authors should clarify the results by first indicating what is statistically significant and then explaining various trends, even if not statistically significant, to avoid misleading the reader during interpretation.

Response: We are extremely grateful to the reviewer for pointing out this problem. we have carefully revised the description of the results. First, we have provided an overview of the general trends observed in hemocyte counts from 6 h to 48 h. Then, we have described the findings that had statistical significance from those that did not (Lines 316-324). Moreover, we have made substantial revisions to the description of the results in sections 3.3 through 3.8 to enhance the clarity and precision of our presentation.

The total number of hemocytes is described as follows:

Compared with the control group, the total number of hemocytes increased from 6 to 18 h and decreased from 24 to 48 h (Figure 3). At 12 h, the total number of hemocytes in the LD20 and LD30 groups was significantly higher than that in the control and LD10 groups (df = 3, 36; F = 25.402; p < 0.0001). At 24 h, the number of hemocytes in the LD20 and LD20 groups was significantly lower than that of the control group (df = 3, 36; F = 23.701; p < 0.0001). At 48 h, the number of hemocytes was significantly lower than that in the control group (df = 3, 36; F = 15.490; p = 0.001). At 6 and 18 h, there was no significant difference in hemocyte counts between the treatment and control groups (df = 3, 36; F = 1.205; p = 0.322; df = 3, 36; F = 0.632; p = 0.599).

10. L410-419: While it is explained that a morphological change in the hemocytes has been observed, it remains uncertain what consequences this may have on physiology and/or immune defense of FAW, especially regarding which hemocytes were affected. Authors could improve their discussion by providing more information on this topic.

Response: Thank you for your professional review work. We have enumerated the morphological damage caused to plasmatocytes and granulocytes by insecticide exposure. This morphological disruption could affect the normal functional capabilities of these hemocytes. Consequently, the insecticides may adversely impact the immune response of insects by impairing the role of these cells. This disruption could potentially weaken the insects' innate resistance to pathogens (Lines 444-456).

The relevant content is as follows:

The sublethal hexaflumuron exposure caused plasmatocyte filopodia to contract and shorten, and granulocytes to compact with a loss of cytoplasmic projections, and changed the morphology and structure of granulocytes in the armyworm Mythimna separata larvae [60]. These studies have shown that chemical exposure can be toxic to hemocyte morphology. The morphology of hemocytes is critical to their function, and changing it can hinder their ability to resist illnesses. For example, granulocytes have been shown to actively produce various sticky nets from their plasma membranes that they use to gather other hemocytes and to implement cellular immune responses [61]. On the other hand, the cytoskeleton of plasmatocytes can rearrange to surround the invading pathogens with pseudopodia or filopodia [10]. In summary, the impairment in the morphology of certain hemocytes can weaken their performance in cellular immunity, resulting in increased susceptibility of insects to pathogens.

The references are as follows:

Huang, Q.; Zhang, L.; Yang, C.; Yun, X.; He, Y. The Competence of Hemocyte Immunity in the Armyworm Mythimna Separata Larvae to Sublethal Hexaflumuron Exposure. Pesti. Biochem. Phys. 2016, 130, 31–38, doi: 10.1016/j.pestbp.2015.12.003.

Cho, Y.; Cho, S. Hemocyte-Hemocyte Adhesion by Granulocytes Is Associated with Cellular Immunity in the Cricket, Gryllus Bimaculatus. Sci. Rep. 2019, 9, 18066, doi: 10.1038/s41598-019-54484-5.

Krendel, M.; Gauthier, N.C. Building the Phagocytic Cup on an Actin Scaffold. Curr. Opin. Cell Biol. 2022, 77, 102112, doi: 10.1016/j.ceb.2022.102112.

11. L420-431: The alteration of THC serves as a crucial starting point for studying the immunotoxic effects of insecticides. However, it is also necessary to explore the potential consequences of insecticide exposure on the different hemocyte classes (DHC) to draw stronger conclusions regarding the potential effects on cellular defense processes driven by various classes of insecticides, such as phagocytosis, encapsulation, and nodulation, which were investigated in this study. Authors could consider to incorporate these aspects to their discussion.

Response: Thanks for your great suggestion on improving the discussion of our manuscript. We added the discussion on the potential impact of insecticides exposure on different hemocyte classes (DHC) and immune defense processes in the revised manuscript (p.14., Lines 461-476).

The relevant content is as follows:

Moreover, different types of hemocytes in different insect species respond differently to stress conditions caused by different insecticides [63]. In the armyworm M. separata larvae, sublethal hexaflumuron exposure decreased granulocytes and increased plasmatocytes, but had few effects on the counts of spherulocytes, oenocytoids, and prohemocytes [60]. The number of prohemocytes and plasmatocytes increased but granulocytes declined in the stingless bee Melipona quadrifasciata exposed to imidacloprid [64]. Among hemocyte types, plasmatocytes and granulocytes are considered major contributors to cellular immunity [5]. The fluctuation in the numbers of these two cell types under insecticide stress may potentially impact cellular immunity. For example, the reduced numbers of granulocytes and plasmatocytes in M. rileyi blastospore-injected FAW may contribute to the compromised capacity of encapsulation and nodulation [14]. However, it is likely that other hemocyte types interact with plasmatocytes and granulocytes, contributing to the overall immune response [65]. The specific dynamics of each type of hemocyte remain unclear and require further investigation.

The references are as follows:

Huang, Q.; Zhang, L.; Yang, C.; Yun, X.; He, Y. The Competence of Hemocyte Immunity in the Armyworm Mythimna Separata Larvae to Sublethal Hexaflumuron Exposure. Pesti. Biochem. Phys. 2016, 130, 31–38, doi: 10.1016/j.pestbp.2015.12.003.

Perveen, N.; Ahmad, M. Toxicity of Some Insecticides to the Haemocytes of Giant Honeybee, Apis Dorsata F. under Laboratory Conditions. Saudi J. Biol. Sci. 2017, 24, 1016–1022, doi: 10.1016/j.sjbs.2016.12.011.

Ravaiano, S.V.; Barbosa, W.F.; Tomé, H.V.V.; Campos, L.A.D.O.; Martins, G.F. Acute and Oral Exposure to Imidacloprid Does Not Affect the Number of Circulating Hemocytes in the Stingless Bee Melipona Quadrifasciata Post Immune Challenge. Pesti. Biochem. Phys. 2018, 152, 24–28, doi: 10.1016/j.pestbp.2018.08.002.

Kwon, H.; Bang, K.; Cho, S. Characterization of the Hemocytes in Larvae of Protaetia Brevitarsis Seulensis: Involvement of Granulocyte-Mediated Phagocytosis. Plos One 2014, 9, e103620, doi: 10.1371/journal.pone.0103620.

12. L472-475: The authors should further discuss in which scenario chlorantraniliprole, by influencing the immunity of FAW, may alter susceptibility to pathogens. For instance, they could explore pathogens commonly found in the field, as well as those used as bio-insecticides, which could potentially modify the insect-pathogens interaction in nature and the IPM programs using different control strategie (e.g., chemical and bio-insecticides).

Response: We express sincere gratitude to the reviewers for raising this issue to improve our discussion section. As suggested by the reviewer, we have described the phenomenon of immune priming, suggesting that FAW may increase resistance to pathogens by enhancing cellular immunity after exposure to chlorantraniliprole. This enhanced immune response could be a contributing factor to the observed resistance in FAW populations. From an IPM perspective, we have suggested using bio-insecticide to weaken the immunity of the FAW before applying chlorantraniliprole to delay the emergence of insecticide resistance (p.15., Lines 526-541).

The relevant content is as follows:

Sheehan et al. [19] expounded a phenomenon known as immune priming, in which prior exposure to a sublethal dose of a pathogen, led to an elevation in the immune response rendering the insect resistant to a subsequent lethal infection a short time later. In our study, sublethal chlorantraniliprole may play a role in immune priming against FAW, suggesting that it may enhance immunity against pathogens. This could be a factor in the development of resistance observed in FAW populations. To address this, the use of microbial pathogens combined with insecticides has been proposed for insect pest management. Sarkhandia et al. [72] reported higher larval mortality when entomopathogenic bacterial strains were combined with chlorantraniliprole and emamectin benzoate to target Spodoptera litura larvae. They speculated that chemical insecticides may act as stressors, weakening the immune response and increasing the susceptibility of insects to bacterial pathogens. From an immunological perspective, we can investigate the effect of chlorantraniliprole in combination with entomopathogenic fungi on FAW. By initially employing pathogenic bacteria to target and weaken the immune system of FAW, we can subsequently apply chlorantraniliprole more effectively. However, additional experiments are necessary to fully explore and validate this method.

The references are as follows:

Sheehan, G.; Farrell, G.; Kavanagh, K. Immune Priming: The Secret Weapon of the Insect World. Virulence 2020, 11, 238–246, doi: 10.1080/21505594.2020.1731137

Sarkhandia, S.; Sharma, G.; Mahajan, R.; Koundal, S.; Kumar, M.; Chadha, P.; Saini, H.S.; Kaur, S. Synergistic and Additive Interactions of Shewanella Sp., Pseudomonas Sp. and Thauera Sp. with Chlorantraniliprole and Emamectin Benzoate for Controlling Spodoptera Litura (Fabricius). Sci. Rep. 2023, 13, 14648, doi: 10.1038/s41598-023-41641-0

Reviewer 2 Report

Comments and Suggestions for Authors

This study reports an impact of chroanthraniliprole on immunity of S. frigiperda. Its sublethal dose enhances the cellular immune responses in THC, nodulation and spreading behavior. To be published, this needs to be reassessed and edited in results.

[Major concerns]

1. Hemocyte classification

    Morphological characters may lead to mis-interpretation in hemocyte classification. I am seriously concerned in cystocyte, which is not clearly defined in hemocyte classification. I recommend to remove Fig 1 to minimize this kind of error-prone interpretation.

2. Delete Fig 9 due to unclear image and little information to support the any conclusion. This should be visualized by a little high resolution confocal microscope.

3. Please add calcium ion experiments

(1) To analyze the insecticide effect on  hemocyte behavior, examine the calcium ion in the hemocytes

(2) Use calcium ionophore or blocker to test the insecticide influnece on hemocytes

Comments on the Quality of English Language

No comment

Author Response

Dear Reviewers,

Thanks for providing us with this great opportunity to submit a revised version of our manuscript entitled "Sublethal chlorantraniliprole exposure enhances cellular immunity in larvae of Spodoptera frugiperda (Smith) (Lepidoptera: Noctuidae)", with the reference number insects-2958906.

We have studied the comments carefully and made major rectifications which we hope meet with approval. And we have made significant improvements to the language used in the manuscript. Below are our detailed responses to the reviewer's comments, highlighted in blue in attachment.

1. Hemocyte classification: Morphological characters may lead to mis-interpretation in hemocyte classification. I am seriously concerned in cystocyte, which is not clearly defined in hemocyte classification. I recommend to remove Fig 1 to minimize this kind of error-prone interpretation. Response: We greatly appreciate the insightful comments provided. Upon thorough consideration of the reviewer's concerns, we acknowledge the potential for misinterpretation regarding the classification of "cystocyte," which lacks a clear definition in the current hemocyte classification. In response to this, we have decided to remove "cystocyte" from Figure 1 in our manuscript to reduce the likelihood of such error-prone interpretations. Additionally, we have carefully reviewed and refined the descriptions of the other five hemocyte types to ensure accuracy and clarity in our study (Lines 274-291). The relevant content is as follows: As shown in Figure 1, five types of hemocytes have been identified in FAW larvae: prohemocytes, plasmatocytes, granulocytes, oenocytoids, and spherulocytes. Of all the cells observed, prohemocytes were small and round with smooth surfaces. The nucleus-to-cytoplasm ratio was high, which helped to differentiate these cells from other types of hemocytes (Figure 1a,b). Plasmatocytes with variable sizes were highly polymorphic, most presenting a spindle shape, while a few showed oval or spherical shapes under the light microscope. These cells were characterized by numerous irregular processes, such as lamellipodia and filopodia. They showed a large centrally localized polymorphic or rounded nucleus (Figure 1c,d). Granulocytes showed a circular or oval profile with highly variable size and contained several refractive inclusions. These cells were characterized by including several dense granules and structured granules. The plasma membrane of these cells was irregular, presenting projections as filopodia surrounding the cell (Figure 1e,f). Oenocytoids appeared as large cells rather regular in shape, with a low nuclear to cytoplasmic ratio. The nucleus of oenocytoids often had an eccentric location. The cytoplasm was homogenous and a few organelles such as vesicles, and small electron-dense granules, could be observed (Figure 1g,h). Spherulocytes were rounded cells and contained a small number of large inclusions (the spherules) that caused the cell to adopt an irregular shape (Figure 1i,j).   Figure 1. Types of hemocytes in FAW larvae. a: The optical microscopy image of Prohemocytes (PR); b: The DIC of Prohemocytes (PR); c: The optical microscopy images of Plasmatocytes (PL); d: The DIC of Plasmatocytes (PL); e: The optical microscopy image of Granulocytes (GR); f: The DIC of Granulocytes (GR); g: The optical microscopy image of Oenocytoids (OE); h: The DIC of Oenocytoids (OE); i: The optical microscopy image of Spherulocytes (SP); j: The DIC of Spherulocytes (SP); Scale bar: 20 μm. 2. Delete Fig 9 due to unclear image and little information to support the any conclusion. This should be visualized by a little high resolution confocal microscope. Response: Thank you for your insightful comments regarding Figure 9. We introduced this figure to provide a more visual representation of the cytoskeletal changes in hemocytes after chlorantraniliprole exposure. We believe that the inclusion of Figure 9 is crucial for a comprehensive understanding of our findings. As suggested by the reviewer, we have taken the following steps to enhance the figure: (a) We have updated Figure 9 with a clearer image to enhance the visual representation of the data. (b) We have added the relevant description in the results section to contextualize the significance of Figure 9 in our study (p.12., Lines 403-406). (c) For the acquisition of the image, we employed the positive fluorescence microscope (Axio Imager Z2, Zeiss, Germany), recognized for its high-resolution imaging capabilities. We acknowledge the superior image clarity achievable with a high-resolution confocal microscope. While we do not currently possess such equipment, we are considering the potential for its use in our future experimental designs. We have attached the revised Figure 9 and relevant content for your review. The relevant content is as follows: The cytoskeleton in the control group was spindle-shaped, and its area was significantly smaller than that of the treatment group (Figure 9A). With the sublethal dose of LD30 chlorantraniliprole exposure, the cytoskeleton extended in all directions and became a rectangle (Figure 9B).         Figure 9. The fluorescent images of cytoskeleton in hemocytes. (A) Control group: the cytoskeleton in hemocytes treated with 0.1% Triton X-100 aqueous solution for 48 h; (B) LD30 group: the cytoskeleton in hemocytes treated with LD30 chlorantraniliprole for 48 h; Scale bar: 100 μm. 3. Please add calcium ion experiments (1) To analyze the insecticide effect on hemocyte behavior, examine the calcium ion in the hemocytes (2) Use calcium ionophore or blocker to test the insecticide influence on hemocytes Response: Thank you for this valuable comment. We carefully considered and evaluated the advice for additional experiments. We fully understand that these experiments would help improve our study, but regrettably, we are temporarily unable to supplement these experiments in a short time. Given the current stage of our research, the additional experiments suggested may take several months to complete. The main purpose of this manuscript is to report a novel phenomenon: sublethal chlorantraniliprole exposure enhances cellular immunity in larvae of Spodoptera frugiperda. To substantiate this discovery, we conducted several experiments about the potential effects of chlorantraniliprole on the abundance, morphology, and function of hemocytes, as well as on some cellular immunity processes to support the discovery and description of this phenomenon. We are confident that even in the absence of the suggested additional experiments, the core discovery of our manuscript remains valid and significant.  In the discussion section, we have expanded upon our findings by integrating relevant literature on the role of Ca2+ ions in hemocyte behavior and cellular immunity within the insect immune system (p.15., Lines 515-526). We acknowledge the value of further research and intend to conduct more in-depth studies in future work to gain a comprehensive understanding of the mechanisms involved. Finally, we greatly appreciate the innovative perspective offered by your suggestion and will certainly consider it for our ongoing and future research.  The relevant content is as follows:  From the insecticide perspective, chlorantraniliprole can induce the release of Ca2+ from the intracellular calcium pool by activating the insect ryanodine receptor (RyR), leading to muscle paralysis and eventual death [36]. Regarding the reports of Ahmed and Kim [69], the inhibition of Ca2+ flux significantly impaired the hemocyte-spreading and subsequent cellular immune response, phagocytosis. They indicated that PGE2 mediates hemocyte-spreading via cAMP signal to activate aquaporin and via Ca2+ signal to activate actin cytoskeletal rearrangement. Therefore, we speculate that chlorantraniliprole may mediate Ca2+ flux in hemocytes, and ultimately affect cytoskeletal rearrangements. However, further studies are needed to investigate the changes in Ca2+ flux in hemocytes after exposure to chlorantraniliprole and the effects of Ca2+ flux on the immunity of FAW larvae. The references are as follows: Lahm, G.P.; Cordova, D.; Barry, J.D. New and Selective Ryanodine Receptor Activators for Insect Control. Bioorgan. Med. Chem. 2009, 17, 4127–4133, doi: 10.1016/j.bmc.2009.01.018. Ahmed, S.; Kim, Y. PGE2 Mediates Hemocyte-Spreading Behavior by Activating Aquaporin via cAMP and Rearranging Actin Cytoskeleton via Ca2+. Dev. Comp. Immunol. 2021, 125, 104230, doi: 10.1016/j.dci.2021.104230.  

Round 2

Reviewer 1 Report

Comments and Suggestions for Authors

General comments:

I would like to thank the authors for their work and for addressing most of my comments. Below, I have outlined some comments where further elaboration from the authors is necessary, along with suggestions for improving language and content throughout the manuscript."

Specific comments to authors' responses:

Authors' response 1:  We sincerely appreciate the valuable comments. As suggested by the reviewer, we have added the Larval Toxicity Bioassay in 2.2 of Materials and Methods (Lines 148-159). We have also added the result of Larval Toxicity Bioassay in 3.1 of Results (p.6., Lines 258-264, Table 1).

REPLY: the larval toxicity bioassay 2.2 (lines 172-184), that might be defined as “baseline toxicity of chlorantraniliprole on FAW larvae”, lacks crucial details. Authors should report (i) the number of FAW larvae tested for each replica, (ii) the laboratory conditions maintained during the bioassays, (iii) the feeding regimen for FAW larvae during the trial, (iv) arena where FAW larvae were maintained after the topical exposure to chlorantraniliprole. Furthermore, it is not clear to me if authors exposed FAW larvae to the following chlorantraniliprole concentrations: 31.25, 62.5, 125, 250, 500 mg/L (line 176), or to 0.25, 0.5, 1, 2, and 4 μg/g (line 170). To ensure clarity and completeness, please provide the missing information and elucidate these aspects.

Authors' response 2: Thank you for the detailed review. We have checked the literature carefully and added the references for the total hemocyte count (p.4., Lines 188), as well as for the phagocytosis (p.5., Lines 201) and encapsulation (p.5., Lines 221).

DONE: I thank the authors for providing required information.

Authors' response 3: Thank you for your comments on our data analysis methodology.

(a) We used the Shapiro-Wilk and Levene's tests to test the assumptions of normality and homogeneity of variance before analysis. For the data to meet the assumptions of normality and homogeneity of variance, we have used one-way ANOVA of variance followed by LSD's multiple comparison analysis for the analysis of significant differences. For the normality assumption that was not met, the non-parametric Kruskal-Wallis test of variance followed by multiple comparison tests have been used for the analysis of significant differences (p.6., Lines 252-264).

REPLY: Thank you for providing information on the data analysis in lines 328-343 and specify you corrected your analysis with the Kruskal Wallis tests when normality assumption was not satisfied. To enhance clarity and readability of the results, it would be beneficial to specify in the statistical analysis section which parameter was tested using ANOVA and which was tested using the Kruskal-Wallis test.  This would provide more clarity and increase the readability of the results.

(b) To show differences among three or more groups, we have used letters instead of "*" to indicate differences between groups in Figure 3-8.

DONE: Thank you for enhancing the figures by incorporating letters following the post-hoc tests.

(c) We have given due consideration to the reviewer's comments to utilize a GLM for our analysis. Our primary focus was on comparing the effects of various doses of chlorantraniliprole on the hemocytes and cellular immunity of the FAW at the same time. Nevertheless, we will take the reviewer's advice into account and incorporate the use of GLM in our future studies to better account for the effects of both dose and time on our experimental results. Figure 3. Effect of chlorantraniliprole on the total hemocyte count in FAW larvae. Data in the figure are presented as mean ± SD. The different treatment groups are compared to the control group at the same time. Different letters on the bars represent significant differences determined by one-way ANOVA of variance and Kruskal–Wallis nonparametric tests (P < 0.05).

REPLY: Thank you for your comment. Regarding Figure 3, it seems the intended abbreviation is "SE" (Standard Error) rather than "SD" (Standard Deviation). I recommend verifying this abbreviation for Figures 4-8 as well. Additionally, could you please clarify the significance of "K" in the legend of each figure, denoting the control (CK)? This clarification will aid in understanding the representation of control groups across all figures.

Authors' response 4: Thank you for the rectification. We have removed this sentence “The hemocytes of the FAW larvae were stained with Giemsa-Wright's stain and then identified by the positive fluorescence microscope.” (p.6., Lines 274).

DONE: authors addressed the request.

Authors' response 5: Thank you for your precious comments. We have now added the degrees of freedom for both between-group and within-group comparisons in sections 3.3 through 3.8. For instance, “at 12 h, the total number of hemocytes in the LD20 and LD30 groups was significantly higher than that in the control and LD10 group (df = 3, 36; F = 25.402; p < 0.0001)”. The notation 'df = 3, 36' indicates that the degrees of freedom are 3 for between-group comparisons and 36 for within-group comparisons, respectively."

REPLY: thank you for providing missing information. Please, specify in line 236 how many FAW larvae did you test for each replicate. Provide the same information for section 2.6, 2.7, 2.8, 2.9, 2.10.

Authors' response 6: We sincerely appreciate the valuable comments. Sublethal chlorantraniliprole exposure severely affected the major morphology of hemocytes in our study, so it was impossible to determine exactly which type of hemocyte morphology was affected. Furthermore, our study examined the morphology of hemocytes at different times after sublethal chlorantraniliprole exposure. However, we found that there were no morphological changes in hemocytes at certain times, so we described the obvious changes at some times in the results (p.7., Lines 300-310).

DONE: thank you for your comment.

Authors' response 7: We apologize for our inaccurate description of the results, we have corrected it (p.8., Lines 317-321). The relevant content is as follows: “At 12 h, the total number of hemocytes in the LD20 and LD30 groups was significantly higher than that in the control and LD10 groups (df = 3, 36; F = 25.402; p < 0.0001). At 24 h, the number of hemocytes in the LD20 and LD20 groups was significantly lower than that of the control group (df = 3, 36; F = 23.701; p < 0.0001)”.

REPLY: thank you for providing corrections in the result section. However, I am not agreeing with the sentences in lines 424-425 “Compared with the control group, the total number of hemocytes increased from 6 to 18 h and decreased from 24 to 48 h (Figure 3).” Indeed, all the treatment are statistically equal at 6h and 18h (figure 3, same letters above bars within the 6h and 18h evaluations). Also, report the correct LD concentration in the sentence in lines 427-429 since “LD20” is repeated twice: ““At 24 h, the number of hemocytes in the LD20 and LD20 groups was significantly lower than that of the control group (df = 3, 36; 428 F = 23.701; p < 0.0001).”

Authors' response 8: Thank you for the suggestion. We have removed “similarly” (Lines 321).

DONE: I thank the authors for providing corrections.

Authors' response 9: We are extremely grateful to the reviewer for pointing out this problem. we have carefully revised the description of the results. First, we have provided an overview of the general trends observed in hemocyte counts from 6 h to 48 h. Then, we have described the findings that had statistical significance from those that did not (Lines 316-324). Moreover, we have made substantial revisions to the description of the results in sections 3.3 through 3.8 to enhance the clarity and precision of our presentation. The total number of hemocytes is described as follows: Compared with the control group, the total number of hemocytes increased from 6 to 18 h and decreased from 24 to 48 h (Figure 3). At 12 h, the total number of hemocytes in the LD20 and LD30 groups was significantly higher than that in the control and LD10 groups (df = 3, 36; F = 25.402; p < 0.0001). At 24 h, the number of hemocytes in the LD20 and LD20 groups was significantly lower than that of the control group (df = 3, 36; F = 23.701; p < 0.0001). At 48 h, the number of hemocytes was significantly lower than that in the control group (df = 3, 36; F = 15.490; p = 0.001). At 6 and 18 h, there was no significant difference in hemocyte counts between the treatment and control groups (df = 3, 36; F = 1.205; p = 0.322; df = 3, 36; F = 0.632; p = 0.599).

REPLY: As pointed out in my comment during the first round of revisions “The authors should clarify the results by first indicating what is statistically significant and then explaining various trends, even if not statistically significant, to avoid misleading the reader during interpretation.”, in my opinion interpreting results based on non-significant differences might not be the most appropriate approach.

As an example, stating in lines 439-441 of section 3.5: "As shown in Figure 4, the phagocytosis ratio was enhanced with sublethal chlorantraniliprole treatment except for LD10 group at 6 h and LD20 group at 12 h and 48 h" may not accurately reflect the statistical significance of the findings. Upon closer examination, it appears that for each time point (6h, 12h, and 48h), all treatments exhibit statistical equality (as indicated by the presence of the same letter above the bars in Figure 4). Hence, it's essential to rely on statistical analysis to discern which LD values are statistically different from the control. I suggest reviewing all results to ensure consistency in this regard.

Inizio modulo

Authors' response 10: Thank you for your professional review work. We have enumerated the morphological damage caused to plasmatocytes and granulocytes by insecticide exposure. This morphological disruption could affect the normal functional capabilities of these hemocytes. Consequently, the insecticides may adversely impact the immune response of insects by impairing the role of these cells. This disruption could potentially weaken the insects' innate resistance to pathogens (Lines 444-456).

The relevant content is as follows:

The sublethal hexaflumuron exposure caused plasmatocyte filopodia to contract and shorten, and granulocytes to compact with a loss of cytoplasmic projections, and changed the morphology and structure of granulocytes in the armyworm Mythimna separata larvae [60]. These studies have shown that chemical exposure can be toxic to hemocyte morphology. The morphology of hemocytes is critical to their function, and changing it can hinder their ability to resist illnesses. For example, granulocytes have been shown to actively produce various sticky nets from their plasma membranes that they use to gather other hemocytes and to implement cellular immune responses [61]. On the other hand, the cytoskeleton of plasmatocytes can rearrange to surround the invading pathogens with pseudopodia or filopodia [10]. In summary, the impairment in the morphology of certain hemocytes can weaken their performance in cellular immunity, resulting in increased susceptibility of insects to pathogens.

DONE: Thank you for addressing my comment.

Authors' response 11: Thanks for your great suggestion on improving the discussion of our manuscript. We added the discussion on the potential impact of insecticides exposure on different hemocyte classes (DHC) and immune defense processes in the revised manuscript (p.14., Lines 461-476).

The relevant content is as follows:

Moreover, different types of hemocytes in different insect species respond differently to stress conditions caused by different insecticides [63]. In the armyworm M. separata larvae, sublethal hexaflumuron exposure decreased granulocytes and increased plasmatocytes, but had few effects on the counts of spherulocytes, oenocytoids, and prohemocytes [60]. The number of prohemocytes and plasmatocytes increased but granulocytes declined in the stingless bee Melipona quadrifasciata exposed to imidacloprid [64]. Among hemocyte types, plasmatocytes and granulocytes are considered major contributors to cellular immunity [5]. The fluctuation in the numbers of these two cell types under insecticide stress may potentially impact cellular immunity. For example, the reduced numbers of granulocytes and plasmatocytes in M. rileyi blastospore-injected FAW may contribute to the compromised capacity of encapsulation and nodulation [14]. However, it is likely that other hemocyte types interact with plasmatocytes and granulocytes, contributing to the overall immune response [65]. The specific dynamics of each type of hemocyte remain unclear and require further investigation.

DONE: Thank you for addressing my comment.

Further minor comments:

Line 175: add “was” before “diluted”.

Line 175: change “series” to “serial”.

Line 177: add “concentration” after “chlorantraniliprole”.

Lines 179-180: Authors stated “The doses of chlorantraniliprole exposed to each larva were…”. Larvae are exposed to each chlorantraniliprole concentrations, not the opposite. Please, correct the sentence to provide more clarity.

Lines 186-187: The following sentence “The sublethal dose or concentration is defined as a dose or concentration that does not cause statistically significant mortality in the experimental population [39]” may lead to confusion. For example, the LD20 tested in this study is a dose that cause 20% of mortality in the experimental population and that affect behavioral and/or physiological traits on individuals that survived to such sublethal pesticide exposure. However, using “does not cause statistically significant mortality” (line 187) could be confusing, as reference 39 reports that “A sublethal dose/concentration is defined as inducing no apparent mortality in the experimental population.”

Line 266: change “test larvae” to “tested larvae”. Please, double check the manuscript for further language errors.

Lines 328-330: it would be helpful to clarify whether the data utilized for probit analysis underwent log transformation or not.

Line 345: Please, specify if dose-mortality relationship was considered valid when there was no significant deviation between the observed and the expected data (at the P > 0.05 level).

Comments on the Quality of English Language

Authors could improve language throught the manuscript. Please, see comments I provided below.

Author Response

Dear Reviewer,

Thanks for providing us with this great opportunity to submit a revised version of our manuscript entitled "Sublethal chlorantraniliprole exposure enhances cellular immunity in larvae of Spodoptera frugiperda (Smith) (Lepidoptera: Noctuidae)", with the reference number insects-2958906.

We would like to extend our sincere gratitude for the reviewer’s comprehensive review and constructive feedback on our manuscript. The reviewer’s insights were invaluable in enhancing the clarity, content, and overall quality of our manuscript. Below, we have addressed the comments that required further elaboration, as well as incorporated the reviewer’s suggestions for language improvements throughout the manuscript, which are highlighted in blue.

Specific comments:

审稿人意见 1:幼虫毒性生物测定 2.2(第 172-184 行)可能被定义为“氯虫苯甲酰胺对 FAW 幼虫的基线毒性”,缺乏关键细节。作者应报告(i)每个复制品测试的FAW幼虫数量,(ii)生物测定期间维持的实验室条件,(iii)试验期间FAW幼虫的喂养方案,(iv)局部暴露于氯虫苯甲酰胺后维持FAW幼虫的场所。此外,我不清楚作者是否将FAW幼虫暴露于以下氯虫苯甲酰胺浓度:31.25、62.5、125、250、500 mg/L(第176行),或0.25、0.5、1、2和4μg/g(第170行)。为确保清晰和完整,请提供缺失的信息并阐明这些方面。

响应:我们衷心感谢您的宝贵意见。我们在手稿中添加了相关细节。

(i) 每次处理3次,每次重复15只幼虫;

(ii)在生物测定过程中保持的实验室条件与昆虫和杀虫剂中描述的相同(27±1°C,70%±5%相对湿度(RH)和10:14h光周期)。我们已经阐明了相关细节(第 166,187 行);

(iii) 试验期间用人工饲料喂养试验幼虫。我们在昆虫和杀虫剂中添加了这一点(第 187 行);

(iv) 所有测试的幼虫,包括局部暴露于氯虫苯甲酰胺的幼虫,均在2.1所述的环境条件下(27±1°C,70%±5%相对湿度(RH)和10:14 h光周期)饲喂。昆虫和杀虫剂。我们在 2.1 和 2.2 部分(第 167,187 行)中添加了此内容。

(v)在幼虫毒性生物测定中,我们采用滴注法处理幼虫。使用该方法分析毒理学数据时,需要根据药物浓度和体积计算每种昆虫的药剂量(ug),并换算每单位昆虫重量(g)的药剂量(ug),其单位为μg/g。我们使用微量注射器将0.2μL氯虫苯甲酰胺(31.25,62.5,125,250,500 mg / L)滴到每个幼虫的肛门(0.025±0.005 g)上,因此暴露于每个幼虫的氯虫苯甲酰胺剂量分别为0,0.25,0.5,1,2,4μg/ g。然后,我们使用0、0.25、0.5、1、2和4 μg/g处理的幼虫的死亡率进行毒理学分析。我们在手稿中澄清了这些细节,以避免混淆(第 181-184 行)。

修改如下:

滴灌法采用Lu等[42]所述,略有修改。简言之,用0.1%Triton X-100(中国成都Beyotime)水溶液将氯虫苯甲酰胺的储备溶液稀释至连续浓度梯度(31.25、62.5、125、250、500mg/L)[40]。使用微量注射器将0.2μL稀释浓度的氯虫苯甲酰胺分别滴入每个四龄幼虫(0.025±0.005g)的肛门上(上海安亭微量进样器厂,上海,中国)。测试幼虫的重量控制在0.025±0.005g。因此,暴露于每个测试幼虫的氯虫苯甲酰胺剂量分别为0.25、0.5、1、2和4μg/g。然后,我们使用用氯虫苯甲酰胺剂量处理的幼虫的死亡率进行分析。对照组 (CK) 用 0.2 μL 0.1% Triton X-100 水溶液处理。每次处理3次,每次重复15只幼虫。试验期间,在上述环境条件下,用人工饲料喂养被试幼虫。在单管中饲养24小时后,记录死亡率数据。当幼虫在用软画笔触摸后仍然不动时,它们被认为是死亡的。

审稿人意见 2:感谢您在第 328-343 行中提供有关数据分析的信息,并说明您在不满足正态性假设时使用 Kruskal Wallis 检验更正了您的分析。为了提高结果的清晰度和可读性,在统计分析部分指定使用方差分析测试的参数和使用 Kruskal-Wallis 检验测试的参数将是有益的。这将提供更清晰的信息,并增加结果的可读性。

响应:我们衷心感谢您的宝贵意见。正如评价员所建议的,我们在图中指定了使用方差分析测试的测试时间,以及使用Kruskal-Wallis测试的测试时间(第449-453行、第474-477行、第505-507行、第531-535行、第552-555行、第580-583行)。

图注(总血细胞计数)中的修改如下:

图3.氯虫苯甲酰胺对FAW幼虫总血细胞计数的影响。图中的数据以平均值±标准差表示,条形上的不同字母表示显著差异(6、12 和 18 小时,单因素方差分析;24 小时和 48 小时,Kruskal-Wallis 非参数检验,p < 0.05)。将 LD10、LD20 和 LD30 组与同一时间点的对照组 (CK) 进行比较。

审稿人意见 3:感谢您的评论。关于图 3,预期的缩写似乎是“SE”(标准误差)而不是“SD”(标准偏差)。我建议也验证图 4-8 的这个缩写。此外,您能否澄清每个图例中“K”的意义,表示控制 (CK)?这一澄清将有助于理解所有数字中对照组的代表性。

响应:对于在稿件中使用“SD”而不是“SE”造成的混淆,我们深表歉意。在审稿人的指导下,我们仔细审查并更新了图3至图8,以准确反映标准误差的“SE”。此外,我们还澄清了名称“CK”,它代表了我们实验中的对照组。具体来说,在我们的测试中,0.1%Triton X-100水溶液作为对照。该说明已纳入材料和方法(第 185 行)的 2.2 和图 3 至 8 的图注释中,以确保读者可以轻松地区分所有相关图中的对照组。

相关内容如下:

对照组 (CK) 用 0.2 μL 0.1% Triton X-100 水溶液处理。每次处理3次,每次重复15只幼虫。

Reviewer's Comment 4: thank you for providing missing information. Please, specify in line 236 how many FAW larvae did you test for each replicate. Provide the same information for section 2.6, 2.7, 2.8, 2.9, 2.10.

Response: We are extremely grateful to the reviewer for pointing out this problem. We have added the number of test larvae per replicate for sections 2.5, 2.6, 2.7, 2.8, 2.9, and 2.10 (Lines 240, 263, 282, 302, 313, and 330).

Reviewer's Comment 5: thank you for providing corrections in the result section. However, I am not agreeing with the sentences in lines 424-425 “Compared with the control group, the total number of hemocytes increased from 6 to 18 h and decreased from 24 to 48 h (Figure 3).” Indeed, all the treatment are statistically equal at 6h and 18h (figure 3, same letters above bars within the 6h and 18h evaluations). Also, report the correct LD concentration in the sentence in lines 427-429 since “LD20” is repeated twice: “At 24 h, the number of hemocytes in the LD20 and LD20 groups was significantly lower than that of the control group (df = 3, 36; 428 F = 23.701; p < 0.0001).”

Response: Thank you for your meticulous review and for pointing out the discrepancies in the result section. We appreciate your attention to detail and acknowledge the errors in the sentences mentioned. We have revised the sentence to accurately represent the data. Furthermore, we have corrected the repetition of "LD20" in the sentence spanning (lines 442-443). The corrected sentence now reads: "At 24 h, the number of hemocytes in the LD10 and LD20 groups was significantly lower than in the control group (df = 3, 36; F = 23.701; p < 0.0001). (df = 3, 36; F = 23.701; p < 0.0001)."

The modifications are as follows:

At 12 h, the total number of hemocytes in the LD20 and LD30 groups increased significantly compared to the control group (df = 3, 36; F = 25.402; p < 0.0001). At 24 h, the number of hemocytes in the LD10 and LD20 groups was significantly lower than in the control group (df = 3, 36; F = 23.701; p < 0.0001). At 48 h, the number of hemocytes in the LD10, LD20, and LD30 groups was significantly lower than that in the control group (df = 3, 36; F = 15.490; p = 0.001). However, at 6 and 18 h, there was no significant difference in hemocyte counts among the LD10, LD20, and LD30 groups and the control group (Figure 3).

Reviewer's Comment 6: As pointed out in my comment during the first round of revisions “The authors should clarify the results by first indicating what is statistically significant and then explaining various trends, even if not statistically significant, to avoid misleading the reader during interpretation.”, in my opinion interpreting results based on non-significant differences might not be the most appropriate approach.

As an example, stating in lines 439-441 of section 3.5: "As shown in Figure 4, the phagocytosis ratio was enhanced with sublethal chlorantraniliprole treatment except for LD10 group at 6 h and LD20 group at 12 h and 48 h" may not accurately reflect the statistical significance of the findings. Upon closer examination, it appears that for each time point (6h, 12h, and 48h), all treatments exhibit statistical equality (as indicated by the presence of the same letter above the bars in Figure 4). Hence, it's essential to rely on statistical analysis to discern which LD values are statistically different from the control. I suggest reviewing all results to ensure consistency in this regard.

Response: Thank you for your valuable guidance during the first round of revisions and for your continued attention to ensuring the accuracy and integrity of our results. Your comment has prompted us to re-evaluate our interpretation of the data. Upon revisiting the results and Figure 4, we acknowledge the oversight in our initial interpretation. As you rightly pointed out, the statement in lines 439-441 of section 3.5 does not accurately convey the statistical significance of the observed trends. We have revised the result section to reflect the correct interpretation based on the statistical analysis (Lines 454-468). Furthermore, we have conducted a thorough review of all results to ensure that our interpretations are consistent with the statistical analyses.

The modifications in the Phagocytosis Assay are as follows:

At 24 h, the phagocytosis ratio was significantly higher in the LD10, LD20, and LD30 groups than in the control group (df = 3, 8; F = 12.434; p < 0.01), with the LD10 group having the greatest effect on promoting phagocytosis ratio. At 18 h, the phagocytosis ratio in the LD10, LD20, and LD30 groups was higher than in the control group and increased with doses, even if there was no significance. At 6, 12, and 48 h, no significant difference was observed in the phagocytosis ratio among the LD10, LD20, and LD30 groups and the control group (Figure 4).

Further minor comments:

Line 175: add “was” before “diluted”.

Response: Thank you for your detailed review. As suggested by the reviewer, we have added "was" before "diluted" (Line 177) and carefully checked the language in the manuscript.

Line 175: change “series” to “serial”.

Response: Thank you for the rectification. We have changed it to "serial" (Line 177).

Line 177: add “concentration” after “chlorantraniliprole”.

Response: Thank you for the detailed review. We have added " concentration" after "chlorantraniliprole" (Line 179).

Lines 179-180: Authors stated “The doses of chlorantraniliprole exposed to each larva were…”. Larvae are exposed to each chlorantraniliprole concentrations, not the opposite. Please, correct the sentence to provide more clarity.

Response: Thanks for your suggestion. We have already stated in the above issues. In the larval toxicity bioassay, we used the drip method to treat the larvae. When using this method to analyze toxicological data, it is necessary to calculate the drug dose (ug) of each insect according to the drug concentration and volume and convert the drug dose (ug) per unit insect weight (g), its unit is μg/g. We used the microsyringe to drop 0.2 µL chlorantraniliprole (31.25, 62.5, 125, 250, 500 mg/L) onto the pronotum of each larva (0.025 ± 0.005 g) so that the dose of chlorantraniliprole exposed to each larva was 0, 0.25, 0.5, 1, 2, 4 μg/g, respectively. Then, we used the mortality of larvae treated at 0, 0.25, 0.5, 1, 2, and 4 μg/g for toxicological analysis. We have clarified these details in the manuscript to avoid confusion (Lines 181-184).

Lines 186-187: The following sentence “The sublethal dose or concentration is defined as a dose or concentration that does not cause statistically significant mortality in the experimental population [39]” may lead to confusion. For example, the LD20 tested in this study is a dose that cause 20% of mortality in the experimental population and that affect behavioral and/or physiological traits on individuals that survived to such sublethal pesticide exposure. However, using “does not cause statistically significant mortality” (line 187) could be confusing, as reference 39 reports that “A sublethal dose/concentration is defined as inducing no apparent mortality in the experimental population.”

Response: Thank you for your insightful observation regarding the potential confusion in the sentence about the definition of a sublethal dose or concentration. We understand that the wording could be misleading, especially in the context of our study where the LD20 does indeed cause a specific percentage of mortality. To clarify and align with the definition provided in reference [39], we have revised the sentence as follows (191-195): “A sublethal dose or concentration refers to a level that induces no apparent mortality in the experimental population, yet may significantly impact behavioral and/or physiological traits in the surviving individuals [39]. In this study, we used the LD10, LD20, and LD30 of chlorantraniliprole to investigate the effects on the survivors' traits post-exposure to the pesticide.” We have made this adjustment in the manuscript to ensure that the definition of a sublethal dose is clear.

The relevant content is as follows:

A sublethal dose or concentration refers to a level that induces no apparent mortality in the experimental population, yet may significantly impact behavioral and/or physiological traits in the surviving individuals [39]. In this study, we used the LD10, LD20, and LD30 of chlorantraniliprole to investigate the effects on the tested larvae of traits post-exposure to the pesticide. After 6 h, 12 h, 18 h, 24 h, and 48 h of treatment, larvae were placed on ice for immobilization. The larvae were sterilized with 75% alcohol, then an insect needle punctured the abdominal prolegs of the larvae. Each larva was collected only once. Hemolymph was transferred to an Eppendorf tube for follow-up experiments.

Line 266: change “test larvae” to “tested larvae”. Please, double check the manuscript for further language errors.

Response: We are extremely grateful to the reviewer for pointing out this problem. As suggested by the reviewer, we have corrected it (Line 269) and carefully checked the manuscript for further language errors.

Lines 328-330: it would be helpful to clarify whether the data utilized for probit analysis underwent log transformation or not.

Response: Thank you for your precious comments. We have clarified the use of log transformation in probit analysis (Line 338-340).

The relevant content is as follows:

In larval toxicity bioassay, the LD10, LD20, LD30, and LD50 values with 95% confidence limits, chi-square value, and slopes of the log-dose probit mortality lines were calculated by probit regression analysis.

Line 345: Please, specify if dose-mortality relationship was considered valid when there was no significant deviation between the observed and the expected data (at the P > 0.05 level).

Response: We sincerely appreciate the valuable comments. In the larval toxicity bioassay, we used probit regression analysis with SPSS 27.0.1 software. In Table 1, the p-value (0.948) represents the significance of Pearson's chi-squared test. The Pearson chi-square test is the degree of fit of the regression line to the observed data. The p-value greater than the 0.05 level indicates that the observed data are close to the expected value of the model, which also indicates that the results of the model are valid and reliable. Thus, the dose-mortality relationship was considered valid. In addition, we clarified what the P-values mean at the bottom of Table 1 (Lines 360-363).

The relevant content is as follows:

LD, CL, χ2, df, and p indicate lethal dose, 95% confidence limits, chi-square value, degrees of freedom, and significance in Pearson's chi-squared test, respectively. The p-value greater than the 0.05 level indicates that the observed data are close to the expected value of the model.

Once again, thank you for your meticulous review and for providing us with the opportunity to improve our work. We hope that the manuscript is now in an acceptable form for publication.
